# Gradient tantalum-doped hematite homojunction photoanode improves both photocurrents and turn-on voltage for solar water splitting

Hemin Zhang [1,4], Dongfeng Li[2,4], Woo Jin Byun [1], Xiuli Wang [2✉], Tae Joo Shin [3], Hu Young Jeong [3✉], Hongxian Han [2], Can Li [2] & Jae Sung Lee[1✉]

Hematite has a great potential as a photoanode for photoelectrochemical (PEC) water splitting by converting solar energy into hydrogen fuels, but the solar-to-hydrogen conversion efficiency of state-of-the-art hematite photoelectrodes are still far below the values required for practical hydrogen production. Here, we report a core-shell formation of gradient tantalum-doped hematite homojunction nanorods by combination of hydrothermal regrowth strategy and hybrid microwave annealing, which enhances the photocurrent density and reduces the turn-on voltage simultaneously. The unusual bi-functional effects originate from the passivation of the surface states and intrinsic built-in electric field by the homojunction formation. The additional driving force provided by the field can effectively suppress charge–carrier recombination both in the bulk and on the surface of hematite, especially at lower potentials. Moreover, the synthesized homojunction shows a remarkable synergy with NiFe(OH)$_x$ cocatalyst with significant additional improvements of photocurrent density and cathodic shift of turn-on voltage. The work has nicely demonstrated multiple collaborative strategies of gradient doping, homojunction formation, and cocatalyst modification, and the concept could shed light on designing and constructing the efficient nanostructures of semiconductor photoelectrodes in the field of solar energy conversion.

[1] School of Energy and Chemical Engineering, Ulsan National Institute of Science and Technology (UNIST), 50 UNIST-gil, Ulsan 44919, Republic of Korea. [2] State Key Laboratory of Catalysis, Dalian Institute of Chemical Physics, Chinese Academy of Sciences, Dalian National Laboratory for Clean Energy, Zhongshan Road 457, 116023 Dalian, China. [3] UNIST Central Research Facilities, Ulsan National Institute of Science and Technology (UNIST), 50 UNIST, Ulsan 44919, Republic of Korea. [4] These authors contributed equally: Hemin Zhang, Dongfeng Li. ✉email: xiuliwang@dicp.ac.cn; hulex@unist.ac.kr; jlee1234@unist.ac.kr

Sunlight is by far the largest source of renewable energy, but its diffuse and intermittent nature slows down widespread utilization[1]. Converting solar energy into storable chemical energy like Mother Nature does in photosynthesis is the most desirable route to move away from the dependence on fossil fuels. Photoelectrochemical (PEC) water splitting stores solar energy into chemical bonds of $H_2$, which is an ideal energy carrier for a completely carbon-free energy economy[2]. Despite intensive research worldwide over the past decades, current state-of-the-art PEC systems do not provide high efficiency and long durability required for practical applications. In fact, over 20% solar-to-hydrogen (STH) conversion efficiencies have already been achieved by using photoelectrodes of high-quality III–V semiconductors[3,4], but their scarcity, high cost of fabrication, and especially lack of long-term stability in aqueous electrolytes are critical obstacles to large-scale applications.

Metal oxide semiconductors are promising photoelectrode materials owing to their availability, low cost, facile synthesis, and excellent oxidative stability[5,6]. In particular, hematite (α-$Fe_2O_3$) is an ideal candidate material in PEC water splitting due to its small band gap (2.1 eV) that allows a high theoretical STH efficiency (15.5%) in addition to the mentioned typical advantages of metal oxides[7]. Notably, a benchmark record of ~6 mA cm$^{-2}$ at 1.23 $V_{RHE}$ under standard conditions was reported by Jeon et al.[8] using combined modifications of hydrogen treatment, ultrathin $TiO_2$ overlayer, and cobalt phosphate cocatalyst loading. Even this superb performance corresponds to <50% of its theoretical limit.

Realizing the high performance of hematite corresponding to its potential remains a huge challenge because of various factors such as small polaron-type charge transport mechanism, short hole diffusion length, poor oxygen-evolution reaction kinetics, and improper conduction band position for unassisted water splitting[9]. Owing to these limitations, the plateau photocurrent density ($J_{ph}$) remains far below the possible 12.4 mA cm$^{-2}$ and the turn-on voltage of photocurrents ($V_{on}$) is generally observed at 0.8–1.0 $V_{RHE}$, which is much more positive than its flat-band potential (~0.45 $V_{RHE}$).

Various modification strategies have been widely employed to improve the PEC performance of hematite, including metallic and nonmetallic doping[10–12], nanostructure engineering[13], heterojunction[14–16], and surface modifications such as passivation overlayers and oxygen-evolution cocatalyst[17–19]. However, every strategy has its own limitations and usually cannot address both $J_{ph}$ and $V_{on}$ simultaneously, the most important two functionalities of the photoelectrode performance. For instance, incorporation of dopants into hematite lattices improves its poor electronic conductivity, but the band bending caused by the dopants also decreases the width of space-charge layer to limit the number of the carriers inside of this layer. Many studies on doping into hematite demonstrate significant increases in $J_{ph}$, but not much effects for reducing $V_{on}$[20,21]. Hematite heterojunction is an effective way to provide efficient charge separation, but it also mainly improves $J_{ph}$ with almost no reduction of $V_{on}$[14,22]. A significant cathodic shift in $V_{on}$ was observed by passivation overlayers of 13-group oxides, but accompanied a reduced $J_{ph}$ value[19]. Only oxygen-evolution cocatalysts (OEC) like cobalt phosphate ("Co-Pi") and $Ni_{1-x}Fe_xOOH$ ($0 \leq x \leq 1$) can improve both $J_{ph}$ and $V_{on}$ simultaneously[17,23], although they improve only surface charge separation and bring ill-defined side effects on light absorption, energy band edges, and character of double layers. Therefore, a strategy is highly desirable that improves both $J_{ph}$ and $V_{on}$ of hematite photoanodes without any undesired side effects.

Our previous reports of hybrid microwave annealing (HMA) took advantage of high temperature annealing in an extremely short time, which showed many distinctive advantages for the synthesis of photoelectrodes such as well-preserved original morphology, high-crystallinity and purity, less-damaged F-doped $SnO_2$ (FTO) conductivity, and inducing unconventional high temperature reactions[22,24–26]. Hydrothermal regrowth method was demonstrated to effectively reduce the structural disorders on or near the surfaces to give a large photovoltage[27]. Herein, we introduce a facile strategy to synthesize gradient tantalum-doped hematite homojunction nanorods by combination of the hydrothermal regrowth and HMA, which exhibits bi-functional effects improving both $J_{ph}$ and $V_{on}$ simultaneously. It was found that the additional driving force provided by the built-in electric field of homojunction significantly improved charge separation both in the bulk and on the surface, especially at lower potentials. Moreover, the synthesized homojunction shows a remarkable synergy with NiFe(OH)$_x$ cocatalyst with significant additional improvements of $J_{ph}$ and $V_{on}$. The displayed outstanding performance by the homojunction-induced built-in electric field was confirmed by surface charge separation efficiency, photoelectrochemical impedance spectroscopy, and transient absorption spectroscopy.

## Results

**Synthesis and characterization of core-shell homojunction hematite nanorods.** The synthesis procedure of hematite homojunction nanorod is shown in Fig. 1. First, in situ tantalum-doped FeOOH (Ta:FeOOH) nanorods were synthesized by a hydrothermal method, and the Ta concentration in the preparation solution was optimized for the desired nanorod morphology (Supplementary Fig. 1) and the best photocurrent–applied potential ($J$–$V$) curve (Supplementary Fig. 2). Then, a thin layer of FeOOH was formed on the as-synthesized Ta:FeOOH nanorods by the hydrothermal regrowth, in which the layer thickness was optimized by the duration (Supplementary Fig. 3). Finally, HMA was employed to convert Ta:FeOOH@FeOOH into Ta:$Fe_2O_3$@$Fe_2O_3$ core-shell homojunction nanorods to take advantage of its unique characteristics: (i) to achieve high temperature of 700–1000 °C in an extremely short time (2–3 min), (ii) to preserve the original nanostructure, and (iii) negligible damage of FTO substrates under these harsh conditions. These distinctive features of HMA are highly desirable for the fabrication of efficient photoelectrodes, which the conventional thermal annealing (CTA) cannot provide[24].

Scanning electron microscopy (SEM) images in Supplementary Fig. 4a, b show the initially synthesized Ta:FeOOH@FeOOH nanorods with diameters of 20–100 nm and lengths of 300–500 nm, in which the FeOOH layer with coarse surface encapsulates the nanorods completely. The Ta:FeOOH nanorods are conformally coated by a thin FeOOH layer, as the initial nanorod

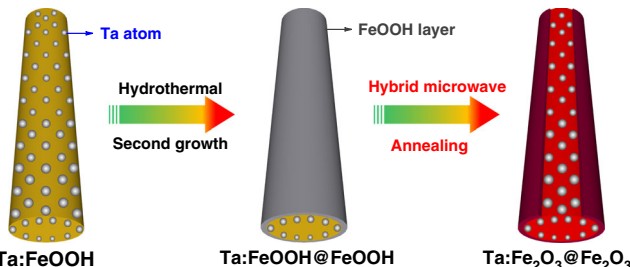

**Fig. 1 Schematic synthesis procedure of Ta:$Fe_2O_3$@$Fe_2O_3$ homojunction nanorods.** Using hybrid microwave annealing, hematite homojunction nanorod is synthesized by hydrothermal regrowth of thin FeOOH layer on Ta:FeOOH nanorods.

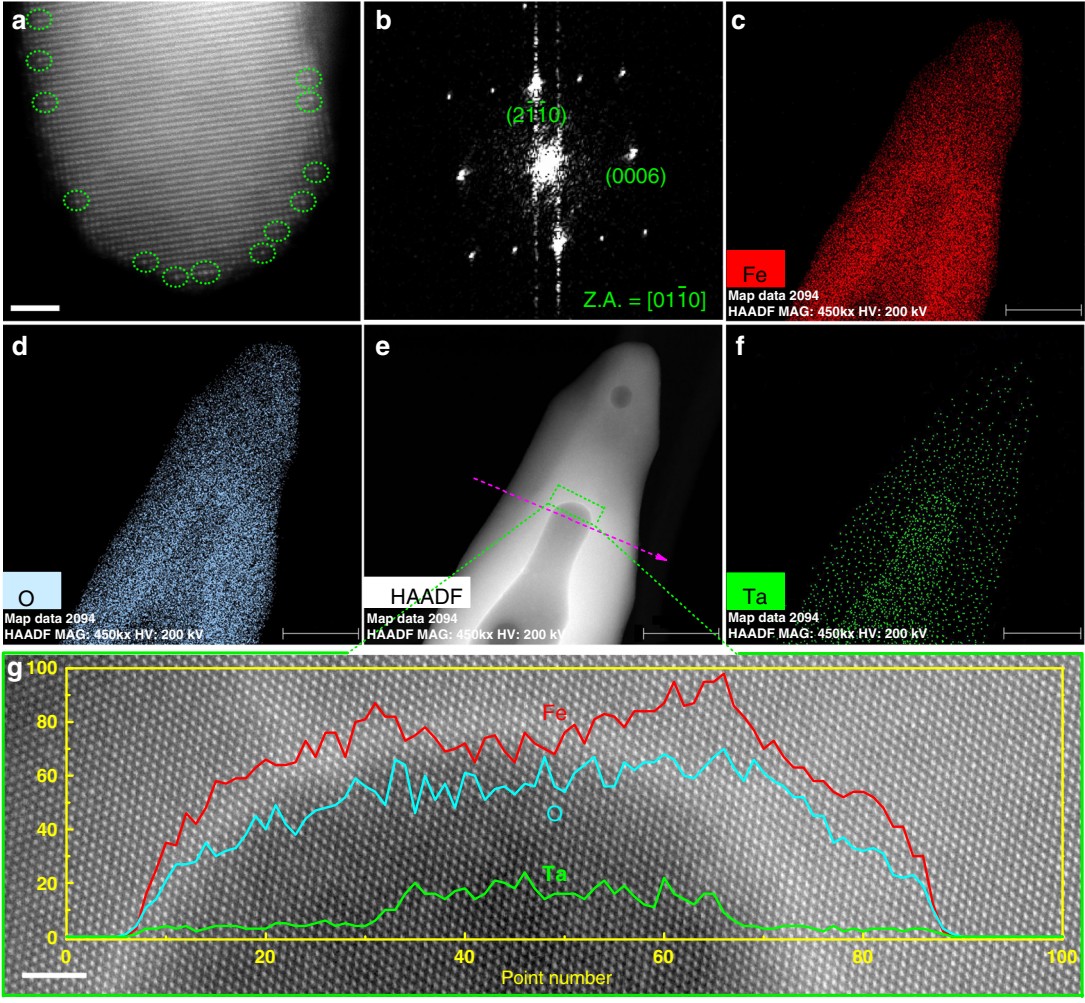

**Fig. 2 HAADF-STEM of Ta:Fe₂O₃ and Ta:Fe₂O₃@Fe₂O₃ homojunction. a** HRSTEM image of Ta:Fe₂O₃ nanorods (circles are Ta atoms). **b** SAED of Ta:Fe₂O₃. **c–f** EELS mapping for a Ta:Fe₂O₃@Fe₂O₃ homojunction nanorod. **g** HRSTEM of homojunction nanorod (inset is the corresponding EELS linear scanning along the dashed arrow in **e**. The scale bar is 40 nm in **c–f** and 2 nm in **a** and **g**.

morphology is well-preserved even after the regrowth (Supplementary Fig. 1e, f). After HMA treatment, however, homojunction nanorods exhibit smooth surfaces (Supplementary Fig. 4c, d), which demonstrates that HMA not only converted Ta:FeOOH@-FeOOH into Ta:Fe₂O₃@Fe₂O₃ phases, but also reconfigured the surface structure by its extremely high temperature annealing in a short time.

The structure of Ta:Fe₂O₃@Fe₂O₃ homojunction nanorods and Ta:Fe₂O₃ were further examined by scanning transmission electron microscopy (STEM). The high-angle annular dark-field STEM (HAADF-STEM) image of Ta:Fe₂O₃ in Fig. 2a (more STEM images in Supplementary Fig. 5) clearly shows Ta atoms in the α-Fe₂O₃ lattices (circled), demonstrating that in situ doping of Ta into FeOOH was successful. Particularly, more Ta atoms appeared near the surface region of α-Fe₂O₃ nanorod, indicating that heavily doped Ta in β-FeOOH nanorods might migrate toward the surface when FeOOH was converted to α-Fe₂O₃ phase under the high temperature of HMA. Besides, the corresponding selective area electron diffraction (SAED) shows the hexagonal structure of single crystal α-Fe₂O₃ with high-crystallinity (Fig. 2b), suggesting that the incorporation of Ta atoms into hematite lattices did not change its intrinsic crystal structure. The electron energy loss spectroscopy (EELS) mapping (Fig. 2f) clearly shows the high Ta concentration in the core and a lower one in the shell, demonstrating clearly the formation of core-shell homojunction.

Notably, hydrothermal regrowth of the nanorods along the axis direction is faster than that along the radial direction because of higher surface energy on the tips of Ta:FeOOH nanorods, leading to a nonuniform conformal shell. Zhu et al. found that the tips of ZnFe₂O₄ nanorods had more efficient hole transport property relative to the bottom part contacting FTO[28]. The nonuniform conformal shell might be expected to reduce this difference, generating a uniform hole transport efficiency for the whole homojunction nanorods. The corresponding high-resolution STEM (HRSTEM) image obtained from the core-shell interface region shows the same perfect crystal lattices without any defects observed (Fig. 2g), indicating the absence of lattice mismatch in homojunction nanorods. Jang et al. reported that the structural disorders on or near surfaces of hematite could be cured by hydrothermal regrowth method[27]. In our case, these structural disorders should have been reconstructed more easily and completely because regrowth of FeOOH takes place on the same phase of Ta:FeOOH instead of on hematite. The linear EELS scanning in Fig. 2g (inset) shows a high concentration of Ta in the core, which accompanies the weakened signals of Fe, suggesting that doped Ta in hematite lattices must be substitutional. In fact, these large metal cation dopants greatly prefer to occupy substitutional rather than interstitial sites[29], which would promote the growth along the {001} basal plane exhibiting stronger (110) diffraction as discussed below[30].

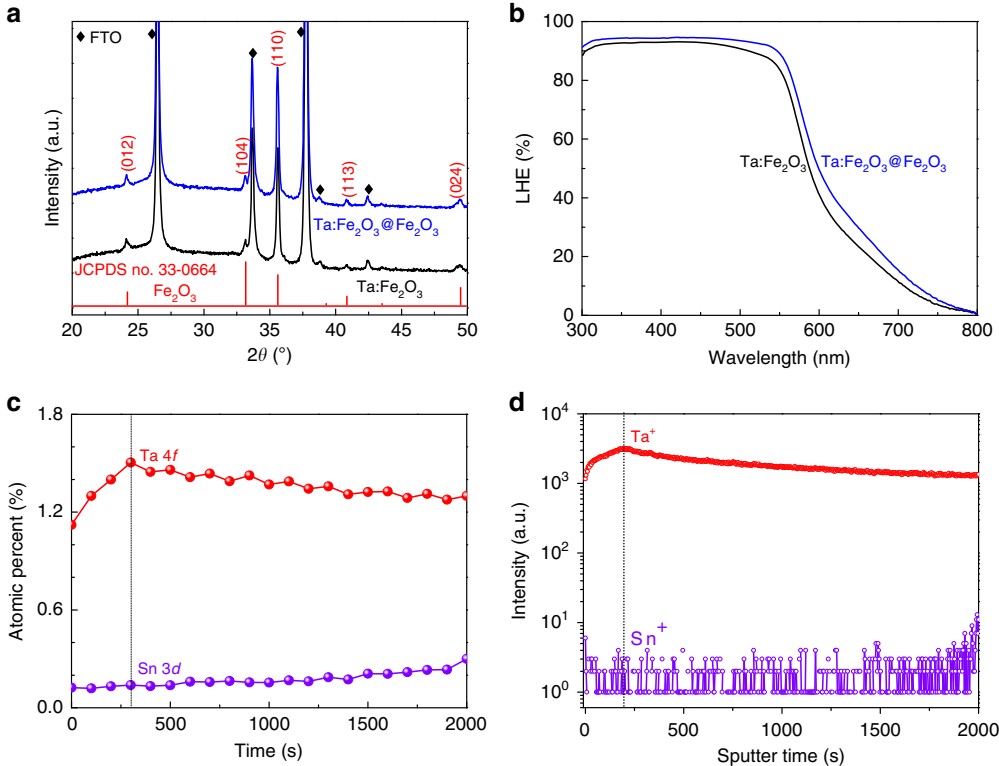

**Fig. 3 Physical characterization of homojunction nanorod. a** XRD and **b** LHE for $Ta:Fe_2O_3$ and $Ta:Fe_2O_3@Fe_2O_3$, and **c** XPS and **d** TOF-SIMS depth profiles for $Ta:Fe_2O_3@Fe_2O_3$.

As discussed in the next section, the $Ta:Fe_2O_3@Fe_2O_3$ homojunction showed greatly enhanced PEC water splitting activity from that of $Ta:Fe_2O_3$. To find origins of the enhancement, X-ray diffraction (XRD) study was carried out first. Both $Ta:Fe_2O_3$ and the homojunction show a very strong (110) peak intensity, indicating the highly preferential crystal orientation (Fig. 3a). Kment et al.[31] demonstrated that the orientation of (110) crystal plane of hematite has a much higher photo-efficiency compared with (104) orientation. However, the two photoanodes show nearly the same XRD behavior (Supplementary Fig. 6), and thus crystal orientation has a negligible contribution to the improved PEC activity. The light harvesting efficiency (LHE, defined as $1-10^{-A}$ where A is absorbance) of the homojunction photoanode shows only a slight enhancement relative to that of the $Ta:Fe_2O_3$ photoanode (Fig. 3b), implying that the influence of light absorption is insignificant as well. The band gap was also almost the same at ~2.1 eV (Supplementary Fig. 7).

Depth profiles of Ta 4f and Sn 3d in the homojunction electrode by X-ray photoelectron spectroscopy (XPS) shows different concentration profiles of Ta and Sn (Supplementary Fig. 8a, b). In Fig. 3c, Ta concentration shows a sharp rising gradient to 1.5% at the etching time of 300 s and then gradually falls, indicating that the core-shell interface is present at the maximum point. On the other hand, Sn concentration from FTO substrate remains very low (only 0.14% at 300 s), showing a gentle, long-range gradient[32]. Time-of-flight secondary ion mass spectrometry (TOF-SIMS) also provides similar elemental distribution in the homojunction (Fig. 3d). The corresponding three-dimensional visualization of SIMS profiles (Supplementary Fig. 9) show the uniform distributions of $Fe^+$ and $Sn^+$, but a clear concentration gradient of $Ta^+$ from top to bottom, although there is no spatial resolution between core and shell parts. Moreover, the calculated shell thicknesses were about 25–30 nm and

30–37.5 nm by the estimated etching rate of XPS (6–7.5 nm $min^{-1}$) and TOF-SIMS (7.5–9 nm $min^{-1}$), respectively (Supplementary Fig. 10). These results of XPS depth profile and TOF-SIMS are consistent with HRSTEM results, demonstrating the formation of gradient Ta-doped hematite homojunction.

To further clarify the state of Ta in the bulk nanorods (isolated atoms or nanoclusters), $Ta:Fe_2O_3$ and $Ta:Fe_2O_3@Fe_2O_3$ were analyzed by synchrotron powder XRD (PXRD) using powder samples scratched from the corresponding photoanode films. Both samples show exactly the same PXRD patterns (Supplementary Fig. 11a), only involving two phases of $Fe_2O_3$ and $SnO_2$ (scratched off from FTO substrate). The Rietveld refinement profile of $Ta:Fe_2O_3@Fe_2O_3$ was obtained by Full Prof Suite software (Supplementary Fig. 11b, c). All the peaks were completely assignable to $Fe_2O_3$ (R-3c) and $SnO_2$ (P 42/mnm). No relevant phases of Ta metal or oxides (such as TaO, $Ta_2O_3$, $TaO_2$ and $Ta_2O_5$) were found. Note that the extreme conditions provided by HMA are more favorable to form high-crystallinity materials relative to CTA. Hence, the nanoclusters (if existed) in the synthesized samples should be crystalline as well, which would have been detected by synchrotron PXRD. Combining synchrotron PXRD with HRSTEM results, we can conclude that the state of Ta in both samples should be isolated and randomly dispersed atoms.

**Photoelectrochemical performance.** The PEC water splitting performance of the optimized $Ta:Fe_2O_3@Fe_2O_3$/FTO photoanode was studied under simulated 1 sun irradiation (100 mW $cm^{-2}$) in 1 M KOH electrolyte in a three-electrode cell with the photo-anode, Ag/AgCl (3 M NaCl), and Pt mesh as working, reference, and counter electrodes, respectively (Supplementary Fig. 12). As shown in Fig. 4a, the $Ta:Fe_2O_3$ photoanode already exhibits a remarkable $J_{ph}$ of 1.93 mA $cm^{-2}$ at 1.23 $V_{RHE}$ due to the effective

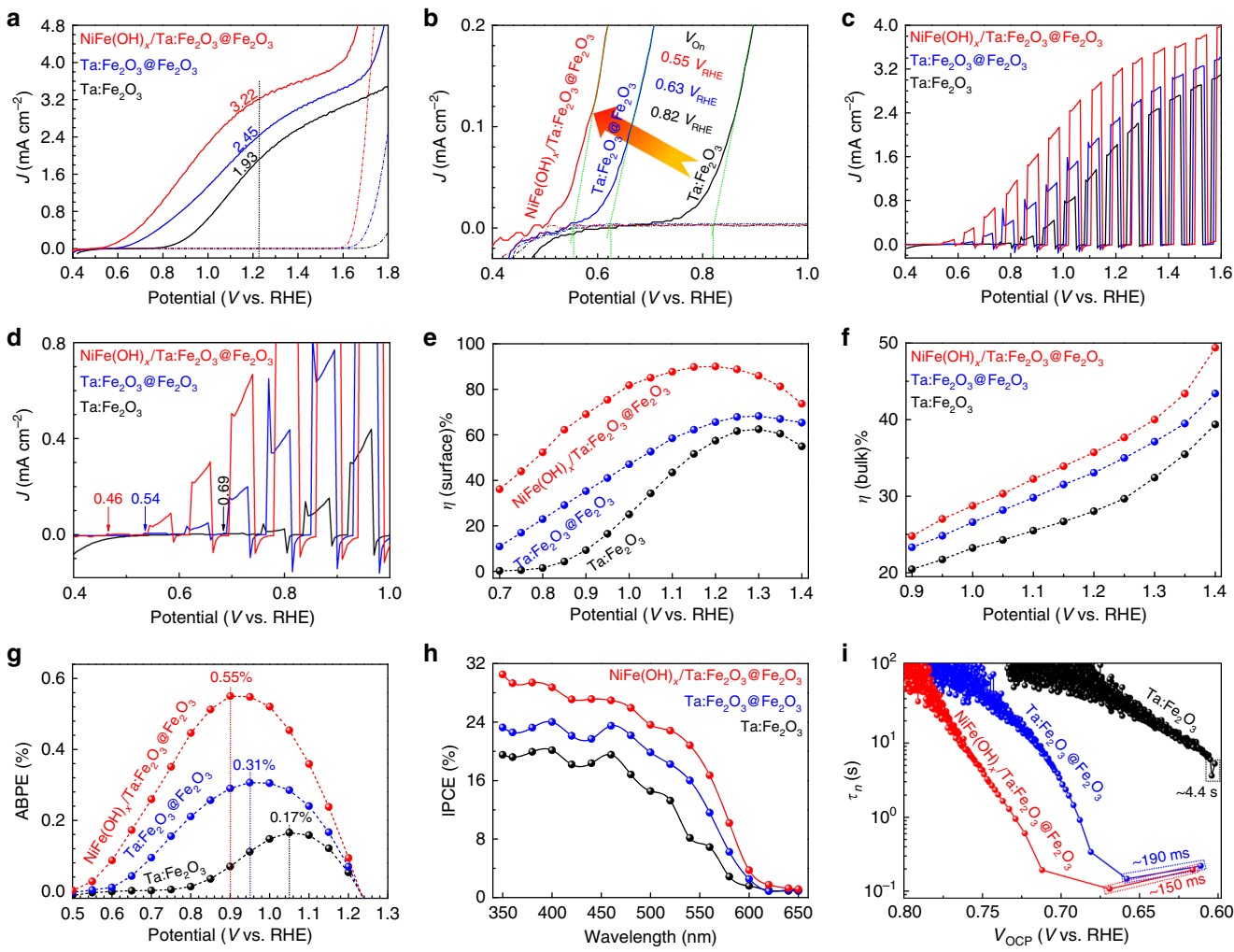

**Fig. 4 Photoelectrochemical performance. a** $J–V$ curves, **b** extracted $V_{on}$, **c** chopped $J–V$ curves, **d** local enlarged chopped $J–V$ curves, **e** surface charge separation, **f** bulk charge separation, **g** ABPE, **h** IPCE, and **i** OCP-derived carrier lifetimes for Ta:$Fe_2O_3$, Ta:$Fe_2O_3$@$Fe_2O_3$, and NiFe(OH)$_x$/Ta:$Fe_2O_3$@$Fe_2O_3$ photoanodes.

Ta doping. When it is modified with a NiFe(OH)$_x$ cocatalyst (Supplementary Fig. 13a), the photocurrent was further improved to 2.48 mA cm$^{-2}$, together with a cathodic $V_{on}$ shift of ~120 mV (Supplementary Fig. 13b). The improvements in both current and $V_{on}$ are typical roles of an efficient cocatalyst. In fact, the performance of Ta:$Fe_2O_3$ is equivalent to the best reported hematite photoanodes doped with various elements (Supplementary Table 1).

The homojunction Ta:$Fe_2O_3$@$Fe_2O_3$ photoanode improved $J_{ph}$ at 1.23 $V_{RHE}$ and $V_{on}$ to 2.45 mA cm$^{-2}$ and 0.63 $V_{RHE}$, respectively (Fig. 5b and Supplementary Fig. 14). It is interesting to note that the effect of homojunction is similar to that of cocatalyst improving both $J_{ph}$ and $V_{on}$ simultaneously. The effect on $J_{ph}$ is very similar, but that on $V_{on}$ shift is much greater for homojunction formation (190 vs. 120 mV). With an additional modification by the NiFe(OH)$_x$ cocatalyst (Supplementary Fig. 15), the homojunction photoanode shows a much higher $J_{ph}$ of 3.22 mA cm$^{-2}$ at 1.23 $V_{RHE}$ and an extremely low $V_{on}$ of 0.55 $V_{RHE}$ (a further cathodic shift by ~80 mV). This performance is superior or comparable to the recently reported state-of-the-art hematite-based photoanodes in terms of the plateau $J_{ph}$, $V_{on}$, and fill factor (Supplementary Table 2). More importantly, NiFe(OH)$_x$ cocatalyst and homojunction formation have synergistic effects each other in improving PEC performance of Ta:$Fe_2O_3$.

It should be noted that all the photoanodes (with/without NiFe(OH)$_x$ overlayer) prepared by conventional thermal annealing (CTA) show very inferior performance (Supplementary Fig. 16), which demonstrates that HMA is indispensable for high efficiency photoanodes. However, even CTA homojunction (Ta:$Fe_2O_3$@$Fe_2O_3$) does exhibit the beneficial effect of built-in electric field (modest $V_{on}$ shift and $J_{ph}$ increase, Supplementary Fig. 16b) relative to bare hematite homojunction ($Fe_2O_3$@$Fe_2O_3$) prepared by either HMA or CTA (only $J_{ph}$ increases, Supplementary Fig. 17), indicating that the core part with a sufficient amount of Ta dopants is prerequisite to construct an effective homojunction so that the generated built-in electric field would promote charge separation efficiently. Interestingly, the superior quality of electrodes prepared by HMA is reflected not only in their own behavior but also in their performance after loading NiFe(OH)$_x$ cocatalyst. This could be related to several issues of CTA, i.e., poor nanorod conductivity due to a limited amount of Sn dopant from FTO, evolved morphology, decreased surface area (Supplementary Fig. 18), and damage of FTO conductivity.

N-type metal ion doping (M$^{n+}$, $n > 3$) is one of the most commonly used methods to enhance the electrical conductivity, which introduces extra electrons near $Fe^{3+}$ sites to form $Fe^{2+}$ sites[33]. The doping increases the polaron (polarized local structure) hopping probability[34], and reduces the effective mass

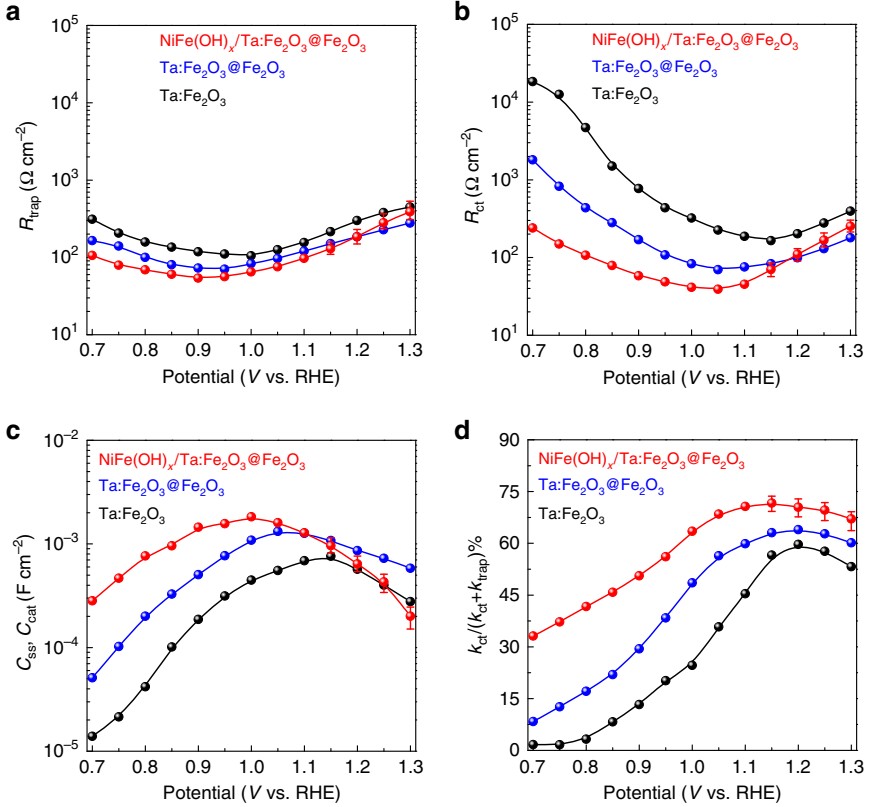

**Fig. 5 PEIS of Ta:Fe$_2$O$_3$, Ta:Fe$_2$O$_3$@Fe$_2$O$_3$, and NiFe(OH)$_x$/Ta:Fe$_2$O$_3$@Fe$_2$O$_3$ photoanodes. a** Trapping resistance. **b** Charge transfer resistance. **c** Surface states capacitance. **d** Ratio of the charge transfer rate constant ($k_{ct}$), and the sum of $k_{ct}$ and trapping rate constant ($k_{trap}$).

of electrons[35], leading to higher $J_{ph}$. However, the doped metal ions in general can provide recombination centers by creating inter-bandgap energy states, especially Fe$^{2+}$ sites near to the surface region (trapping states), which could increase the overpotential for water oxidation[36,37].

The formation of gradient Ta-doped homojunction provides several advantages. First, the reduced density of trapping states by less defective surfaces (fewer Fe$^{2+}$ sites induced by Ta$^{5+}$ dopants) are beneficial to decrease its $V_{on}$, which is directly reflected by the lower flat-band potential (Supplementary Fig. 19). Second, the extracted photovoltage increases by ~110 mV with homojunction formation and further by ~40 mV by NiFe(OH)$_x$ cocatalyst modification (Supplementary Fig. 20). The generated big photovoltage by the homojunction suggests the presence of an intrinsic built-in electric field[38]. Third, the preservation of high electrical conductivity in the bulk (by high concentration Ta$^{5+}$ dopants) promotes the rapid separation of photogenerated charge carriers. Besides, n$^+$–n homojunction by heavily and slightly doped Ta$^{5+}$ in the core and shell parts provides a supplemental charge separation by an increased built-in electric field[39]. Finally, the gradient Ta dopants benefits the smooth flow of the separated charge carriers as schematically shown in Supplementary Fig. 21. Abdi et al.[40] demonstrated that the gradient doped n$^+$–n homojunction could induce less recombination, improve spatial distribution of the electric field, and enlarge the space-charge region relative to that of homojunction without gradient dopant, resulting in a more efficient charge separation.

From the chopped $J$–$V$ curves (Fig. 4c), homojunction shows an increased transient photocurrent both under the light and in dark relative to that of Ta:Fe$_2$O$_3$, suggesting the capable extraction of photoholes onto the surface. In a sense, the shell part of homojunction provides a hole-storage layer to promote sluggish water oxidation[41], which on the other hand limits the

improvement of $J_{ph}$ owing to the enough-time-induced high possibility of recombination and back reaction (oxygen reduction)[42]. By a cocatalyst of NiFe(OH)$_x$, the transient photocurrents declines greatly, confirming the role of cocatalyst to efficiently transfer the accumulated charge carriers into the reaction of water oxidation. The enlarged part of chopped $J$–$V$ curves (Fig. 4d) shows a much lower turn-on voltage of photoresponse at 0.46 V$_{RHE}$ for cocatalyst-modified homojunction (0.54 and 0.69 V$_{RHE}$ for homojunction and Ta:Fe$_2$O$_3$, respectively), which almost approaches the theoretical value of hematite (0.45 V$_{RHE}$)[27].

Bulk ($\eta_{bulk}$) and surface ($\eta_{surface}$) charge separation efficiencies were determined in the same electrolyte with addition of 0.5 M H$_2$O$_2$ as a hole scavenger as described in Supplementary Fig. 22 and Supplementary Note section. The $\eta_{bulk}$ denotes the fraction of holes that reach the electrode/electrolyte interface without recombination in the bulk of electrode, while $\eta_{surface}$ represents the fraction of those holes at the interface that are successfully injected to the water oxidation. In Fig. 4e, $\eta_{surface}$ of Ta:Fe$_2$O$_3$ already reaches 61.6% at 1.25 V$_{RHE}$, while the homojunction improves it modestly to 67.9%. Interestingly, however, the homojunction offers much more significant improvements at a lower potential range from 0.7 to 1.1 V$_{RHE}$. For example, $\eta_{surface}$ of the homojunction increases by almost 7- and 4-folds from those of Ta:Fe$_2$O$_3$ at 0.85 and 0.90 V$_{RHE}$, respectively. This behavior should be related to its intrinsic built-in electric field, which provides a supplemental driving force to separate charge carriers, essential especially at lower applied potentials. The lowered $V_{on}$ (Fig. 4b) can be interpreted by the same reason. In stark contrast, $\eta_{surface}$ of cocatalyst-modified homojunction shows a remarkable enhancement in the whole potential range, which finally reaches 88.7% at 1.20 V$_{RHE}$. Altogether with the suppressed transient current in Fig. 4c, the result indicates that the cocatalyst passivates many surface states present on the surface of

homojunction. In Fig. 4f, Ta:Fe$_2$O$_3$ shows respectable $\eta_{bulk}$ of 29.6% at 1.25 V$_{RHE}$, ~2-fold higher than the previous report of bare hematite[22,26], indicating a significant effect of gradient Ta doping. Formation of homojunction further increases $\eta_{bulk}$ to 35.1% at 1.25 V$_{RHE}$, indicating that the homojunction assists bulk charge separation as well. However, cocatalyst of NiFe(OH)$_x$ has nearly no contribution to $\eta_{bulk}$ because its effect remains only on the surface.

The applied-bias photon-to-current conversion efficiency (ABPE = $J_{ph}$ (1.23–$E$)/$P_{irradiation}$) represents the net effect of photoelectrode by subtracting the contribution of bias potential ($E$). Fig. 4g shows increasing peak ABPE values in order of Ta:Fe$_2$O$_3$ (0.17% at 1.05 V$_{RHE}$) < Ta:Fe$_2$O$_3$@Fe$_2$O$_3$ (0.31% at 0.95 V$_{RHE}$) < NiFe(OH)$_x$/Ta:Fe$_2$O$_3$@Fe$_2$O$_3$ (0.55% at 0.90 V$_{RHE}$). Note also that formation of homojunction and further cocatalyst modification lower the applied-bias to reach its maximum by ~100 and ~50 mV, respectively, which is in line with the corresponding $V_{on}$ shifts (Fig. 4b). The incident photon-to-current conversion efficiency (IPCE) in Fig. 4h is also significantly enhanced by formation of homojunction and further cocatalyst modification over the entire photon energy (340–600 nm). IPCE values of all samples approach zero at wavelengths beyond 600 nm, in agreement with the band gap of hematite. As mentioned earlier, there is no significant difference of light harvesting capability between Ta:Fe$_2$O$_3$ and the homojunction (Fig. 4b). NiFe(OH)$_x$ cocatalyst on the homojunction does not change any light absorption because it is extremely thin (Supplementary Fig. 15b, e). Taken all these together, the enhanced IPCE by homojunction and cocatalyst is due to the improved charge separation and passivation of surface states, respectively. Moreover, the integrated IPCE with reference to AM 1.5G spectrum at 1.23 V$_{RHE}$ gives a solar photocurrent ($J_{SC}$) of 1.86, 2.38, and 3.07 mA cm$^{-2}$ for Ta:Fe$_2$O$_3$, Ta:Fe$_2$O$_3$@Fe$_2$O$_3$, and NiFe(OH)$_x$/Ta:Fe$_2$O$_3$@Fe$_2$O$_3$, respectively (Supplementary Fig. 23), which are very close to the experimentally measured $J_{ph}$ at 1.23 V$_{RHE}$ (Fig. 4a).

The open circuit potential (OCP) transient decay profile can provide additional information on built-in electric field and its effects on the behavior of generated charges. Compared with Ta:Fe$_2$O$_3$, the homojunction photoanode shows a remarkably accelerated OCP decay, indicative of a larger photovoltage generation ($\Delta$OCP = OCP$_{dark}$ – OCP$_{light}$) (Supplementary Fig. 24), which demonstrates that homojunction can provide an additional intrinsic built-in electric field to enhance the driving force of charge separation. The OCP decay is further enhanced by loading NiFe(OH)$_x$ cocatalyst on the homojunction, confirming the effectiveness of cocatalyst modification. For comparison of charge recombination rate at the photoanode/electrolyte junction, the carrier lifetime was quantified by[43]:

$$\tau_n = -\frac{\kappa_B T}{e}\left(\frac{dOCP}{dt}\right)^{-1} \quad (1)$$

where $\tau_n$, $k_B$, $T$, $e$, and dOCP/d$t$ are the carrier lifetime, Boltzmann's constant, temperature (K), charge of single electron and derivative of the OCP transient decay, respectively. The carrier lifetime of homojunction is ~190 ms at the transient when illumination is stopped (Fig. 4i), which is smaller by a factor of 23 compared to ~4.4 s for Ta:Fe$_2$O$_3$. The fast decay kinetics is indicative of the enhanced charge recombination when illumination is removed, which suggests on the other hand that the charge trapping is insignificant when illuminated[16,44]. Consequently, the effective charge separation is expected in the homojunction photoanode under PEC water splitting condition.

In a 5 h stability test (Supplementary Fig. 25), Ta:Fe$_2$O$_3$ loses some of its stability by homojunction formation (96.5% → 92.8%)

owing to a small amount of Ta on the surface with a stronger bond energy (Ta–O>>Fe–O)[45]. With the cocatalyst modification, however, homojunction stability is improved up to 99.2%, significantly better than the pristine Ta:Fe$_2$O$_3$. The gases evolved from the optimized photoanode (NiFe(OH)$_x$/Ta:Fe$_2$O$_3$@Fe$_2$O$_3$) and the Pt counter electrode were quantified by gas chromatography, which was held under the standard conditions for 200 min (Supplementary Fig. 26). The ratio of evolved O$_2$ and H$_2$ is close to 1:2, and the Faraday efficiencies of O$_2$ and H$_2$ evolution reaction are 93% and 96%, respectively (Supplementary Fig. 27). The results demonstrate that the measured $J_{ph}$ is indeed due to O$_2$ and H$_2$ evolution reaction from PEC water splitting.

**Photoelectrochemical impedance spectroscopy (PEIS).** A representative two-RC-unit equivalent circuit (Supplementary Fig. 28a) is used to fit the Nyquist plots of PEIS (Supplementary Fig. 29), where the central role of surface states is highlighted in oxygen-evolution reaction[46]. For cocatalyst-modified homojunction, however, PEIS spectra show a different behavior. A third semicircle in Nyquist plots above 1.1 V$_{RHE}$ appears at low frequencies (Supplementary Fig. 29c), which requires another RC unit to fit its equivalent circuit (Supplementary Fig. 28b). In this case, $C_{ss}$ should be replaced with $C_{cat}$ because the charge transfer takes place mainly through the cocatalyst layer[47]. Bode plots of all the samples display corresponding variations (Supplementary Fig. 30). To justify the validity of PEIS and fitting results, the total resistance ($R_{tot} = R_s + R_{trap} + R_{ct}$) of each sample is plotted against the differential resistance ($R'_{tot} = dV/dJ$) from the measured $J$–$V$ curves. The comparison shows that the dotted $R_{tot}$ matches the derivative curve of $R'_{tot}$ reasonably well for each sample (Supplementary Fig. 31).

Fig. 5a, b shows that $R_{trap}$ and $R_{ct}$ values of the NiFe(OH)$_x$/Ta:Fe$_2$O$_3$@Fe$_2$O$_3$ photoanode are lower than those of Ta:Fe$_2$O$_3$ and Ta:Fe$_2$O$_3$@Fe$_2$O$_3$ in the potential range of 0.7–1.2 V$_{RHE}$. The $R_{trap}$ values show only slight variations, but $R_{ct}$ decreases significantly by formation of homojunction, and further by cocatalyst modification, which represents the trend of markedly increasing $J_{ph}$ by the successive modifications. Notably, the potential of minimal $R_{ct}$ value for homojunction is smaller than that of Ta:Fe$_2$O$_3$ by ~100 mV, which corroborates its much earlier $V_{on}$ (~190 mV negatively shifted) again. The additional cocatalyst modification shows a modest shift of this potential, consistent with the smaller $V_{on}$ shift (~80 mV). $C_{ss}$ corresponding to the surface states shows a Gaussian distribution (Fig. 5c), which results from the activation of intermediate species (Fe$^{IV}$ = O) in oxygen-evolution reaction[48]. The $C_{ss}$ peak of the homojunction shows a lower potential by ~100 mV than that of Ta:Fe$_2$O$_3$. Additional cocatalyst modification further decreases the potential of $C_{ss}$ peak by ~50 mV. As proposed by Hamann and coworkers[46], positions of the $C_{ss}$ peak and the dip in $R_{ct}$ profiles correlate with generated photocurrents since the charge transfer to the electrolyte goes through the surface states. The results also agree well with the $J$–$V$ curves in Fig. 4a.

Charge transfer and surface recombination through surface states are two competing processes that determine the rate of water oxidation on hematite photoanode surface. Assuming that the rates of charge transfer and trapping are inversely proportional to their respective resistances, the charge transfer efficiency in these two competing processes is represented as[49]:

$$\text{Transfer efficiency}(\%) = \frac{k_{ct}}{k_{ct} + k_{trap}} = \frac{R_{trap}}{R_{ct} + R_{trap}} \quad (2)$$

where $k_{ct}$ and $k_{trap}$ are the rate constants of charge transfer and trapping, respectively. Fig. 5d shows the charge transfer efficiency at different potentials. The maximum charge transfer efficiency

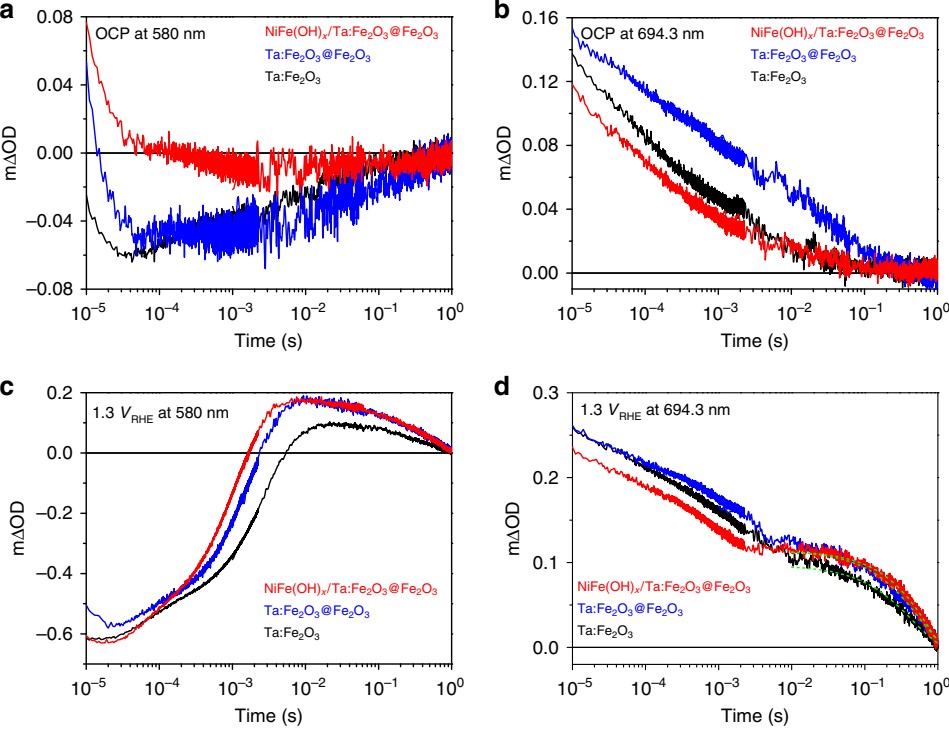

**Fig. 6 Transient absorption spectroscopy (TAS) for Ta:Fe₂O₃, Ta:Fe₂O₃@Fe₂O₃, and NiFe(OH)ₓ/Ta:Fe₂O₃@Fe₂O₃ photoanodes. a** OCP at 580 nm. **b** OCP at 694.3 nm. **c** 1.3 $V_{RHE}$ at 580 nm. **d** 1.3 $V_{RHE}$ at 694.3 nm.

obtained at nearly the same potential (1.15–1.20 $V_{RHE}$) is 59.6%, 64.1%, and 71.4% for Ta:Fe₂O₃, Ta:Fe₂O₃@Fe₂O₃, and NiFe (OH)ₓ/Ta:Fe₂O₃@Fe₂O₃, respectively, which is quite close to its corresponding $\eta_{surface}$ (61.6%, 67.9%, and 88.7%, respectively). The mutually verifying results further demonstrate that homojunction can effectively suppress charge recombination (improve charge transfer), especially at lower potentials, where additional driving force provided by its intrinsic built-in electric field has a more pronounced effect. However, the profiles of $\eta_{surface}$ (Fig. 4e) and transfer efficiency (Fig. 5d) show some discrepancy probably caused by partial loss of charge transfer, the existence of direct charge transfer from the valence band, and imperfect equivalent circuit fitting[50]. In the case of NiFe(OH)ₓ/Ta:Fe₂O₃@Fe₂O₃, the charge transfer becomes complicated involving a dominant path through the cocatalyst layer, few surface states (mostly passivated), and direct transfer from the valence band, which cause a little bigger discrepancy in comparison with the other two photoanodes.

**Transient absorption spectroscopy (TAS).** TAS was used here to further understand the effects of homojunction and additional cocatalyst modification. The distinct absorption spectra of photoelectrons and holes in the photoelectrodes make it possible to monitor the concentration variation of these species under *operando* conditions[51,52]. TAS of hematite typically displays two bands: a bleach band (decreased optical density) at a probe wavelength of ~580 nm is attributed to trapping photoelectrons, and an absorption band at >650 nm is assigned to photoholes[52–54]. Hole scavenger reagent of Na₂SO₃ was chosen to confirm the TAS signal (Supplementary Fig. 32). As expected, Na₂SO₃ consumes photoholes evidenced by decreased amplitude of 694.3 nm absorption, resulting in more photoelectrons revealed by the increased amplitude of 580 nm.

Under OCP condition where the electrodes were immersed in 1 M NaOH electrolyte without external bias (Fig. 6a, b and

Supplementary Fig. 33a), the photogenerated charge–carrier dynamics in hematite photoanodes should be assigned to its bulk electron–hole recombination processes, since their decay dynamics are faster than the timescale of water oxidation on hematite surface as shown in bias-dependent TAS dynamics (Fig. 6d). Nevertheless, the existence of semiconductor/ liquid junction will affect the recombination processes of charge/ carrier relative to that in air (Supplementary Fig. 34). Compared with Ta:Fe₂O₃ photoanode, TAS of homojunction photoanode shows a slightly reduced bleach amplitude (Fig. 6a), indicating some passivation of electron trap states[54]. More importantly, the homojunction photoanode shows a remarkable retardation of the fast decay curves and results in a long half-lifetime ($\tau_{50\%}$ = 1.4 ms) at the 694.3 nm transient relative to Ta:Fe₂O₃ ($\tau_{50\%}$ = 0.23 ms), demonstrating that homojunction greatly suppresses charge–carrier recombination and generates a larger amount of residual photoholes. These results further confirm the existence of built-in electric field between the core (Ta:Fe₂O₃) and the shell (Fe₂O₃) parts of homojunction with the field direction from inside to outside, which significantly promotes photogenerated charge separation and accumulation. For the NiFe(OH)ₓ modified homojunction, the bleach amplitude at 580 nm decreases dramatically relative to the bare homojunction photoanode, which suggests the large decrease of trapped photoelectrons by passivation of surface electron trap states[54]. On the other hand, the absorption amplitude at 694.3 nm also slightly decreases and the half-lifetime decreases to 0.19 ms due to the reduction of electron trap states, resulting from the increase of direct recombination[55] or the charge transfer from homojunction to NiFe(OH)ₓ[56].

To understand the actual PEC processes, *operando* TAS experiments were implemented under an applied-bias of 1.3 $V_{RHE}$ (Fig. 6c, d and Supplementary Fig. 33b). The bleach recovery of Ta:Fe₂O₃@Fe₂O₃ ($\tau_{50\%}$ = 0.90 ms) and NiFe(OH)ₓ/ Ta:Fe₂O₃@Fe₂O₃ ($\tau_{50\%}$ = 0.51 ms) photoanodes is much faster than that of Ta:Fe₂O₃ ($\tau_{50\%}$ = 1.3 ms), which illustrates that the

trapped electrons were extracted more efficiently. In terms of the absorption decay dynamics at 694.3 nm (1.3 $V_{RHE}$), three photoanodes exhibit similar biphasic decay, including electron–hole recombination and water oxidation: a fast decay phase (<10 ms) assigned to electron–hole recombination at early times, and a slow decay phase (10 ms–s) assigned to the population of long-lived photoholes for water oxidation. Ta: $Fe_2O_3@Fe_2O_3$ (5.5 ms) shows a much shorter time relative to that of Ta:$Fe_2O_3$ (11 ms) at the turning point of the fast and slow phases (Supplementary Fig. 35), indicating that homojunction formation can efficiently employ photoholes at space-charge region from the view of timescale. In addition, bias-dependent TAS results in Supplementary Fig. 36 display that the long-lived holes appear at lower bias (0.7–0.8 $V_{RHE}$) in homojunction photoanode than that in Ta:$Fe_2O_3$ photoanode (0.8–0.9 $V_{RHE}$), which agrees well with the negative shift of onset potential. Therefore, the homojunction formation with $Fe_2O_3$ surface is beneficial to take advantage of photoholes at the space-charge region (Supplementary Fig. 37). Moreover, NiFe(OH)$_x$ modification can further shorten the time to achieve turning point (2.3 ms), revealing a promotional effect in surface reaction dynamics. By mono-exponential fitting for timescale of 10 ms–1 s, homojunction samples with/without NiFe(OH)$_x$ modification display more long-lived holes, which will facilitate the PEC performance.

## Discussion

Gradient tantalum-doped hematite homojunction photoanode prepared by the hydrothermal regrowth and HMA methods exhibits a distinctively bi-functional effect, enhancing photocurrent density and reducing turn-on voltage $V_{on}$ of the photocurrents simultaneously. The additional driving force provided by the built-in electric field of the homojunction effectively improves charge–carrier separation, which originates from the different amounts of tantalum dopants between the core and shell parts of the homojunction. Its effects were particularly more significant at lower potentials. Moreover, the synthesized homojunction shows a remarkable synergy with NiFe(OH)$_x$ cocatalyst with significant additional improvements of $J_{ph}$ and cathodic shift of $V_{on}$. As a result, the finally optimized homojunction photoanode (including dopant amount, layer thickness and cocatalyst) improves $J_{ph}$ by 66.8% from 1.93 to 3.22 mA cm$^{-2}$ at 1.23 $V_{RHE}$, and cathodic shift $V_{on}$ by ~270 mV from 0.82 to 0.55 $V_{RHE}$ under standard PEC water splitting conditions.

Metal doping into hematite is quite essential to improve its intrinsically poor conductivity. However, it accompanies increased surface defects and changed coordination of iron that bring a high population of surface states on hematite, which are known to cause Fermi level pinning and increase trap-mediated recombination. This study has demonstrated that homojunction formation can passivate the surface states and provide an electric field as an additional driving force to suppress charge recombination, leading to cathodic shift of $V_{on}$ and enhanced $J_{ph}$. It should be noted that HMA-induced high temperature can also anneal one of two surface states identified by Hamman and coworkers[57], leading to reduction of surface recombination and Fermi level pinning. Consequently, homojunction enables NiFe(OH)$_x$ cocatalyst to perform a much better than that on Ta:$Fe_2O_3$ photoanode, indicating that successive modifications of high doping, homojunction, and cocatalyst loading work synergistically to improves the PEC water splitting performance of hematite photoanodes providing high $J_{ph}$ and low $V_{on}$. This general concept of multiple modifications could be extended to other metal oxide semiconductors to boost their PEC activities.

## Methods

**Fabrication of Ta:FeOOH@FeOOH nanorods on F:SnO$_2$ (FTO) substrate.** Several pieces of conductive FTO substrates (PECTM 8, 6–9 Ω, Pilkington) of 25 × 50 mm$^2$ were ultrasonically cleaned for 30 min in detergent (deconex® 11 Universal), ethanol, and acetone to make a sufficiently hydrophilic surface. The Ta: FeOOH nanorods grew on FTO at 100 °C for 2 h using 100 mL of an aquous solution of 4.0 g FeCl$_3$·6H$_2$O (Sigma Aldrich, ≥99%), 200 μL HNO$_3$ (Sigma Aldrich, 70%), 8.4 g NaNO$_3$ (Sigma Aldrich, ≥99%), and different amounts of TaCl$_5$ solution (Sigma Aldrich, 99.8%). The obtained yellow-colored Ta:FeOOH nanorod films were rinsed with abundant deionized water. Subsequently, the second growth of pure FeOOH layer on the surface of Ta:FeOOH nanorods was conducted at 90 °C for various durations (15, 30, and 60 min) in the same solution but without TaCl$_5$. Finally, the fabricated Ta:FeOOH@FeOOH nanorods on FTO was sufficiently rinsed by deionized water.

**Homojunction core-shell Ta:Fe$_2$O$_3$@Fe$_2$O$_3$ photoanode by HMA.** The well-cut samples (12.5 × 12.5 mm$^2$) were put on the slightly compacted graphite powder (60 mL) as a susceptor in a Pyrex beaker (100 mL) and treated in a home-made microwave oven (2.45 GHz, 1000 W) for 2 min at the full power. After HMA, the Ta:FeOOH@FeOOH was converted into Ta:Fe$_2$O$_3$@Fe$_2$O$_3$ without significant change of morphology.

For cocatalyst modification, a facile, one-step precipitating despostion method was used to prepare amorphous metal (oxy)hydroxide on the surface of nanorods[25]. Typically, the fabricated photoelectrode was immersed into a solution containing 5 × 10$^{-3}$ M FeCl$_3$·6H$_2$O and NiCl$_2$·6H$_2$O (Sigma Aldrich, ≥99%) for different durations (1, 2, and 3 h) at 30 °C. Subsequently, the photoelectrode was rinsed by sufficient deionized water.

**Physical characterization.** X-ray diffraction (XRD) spectra were obtained by PW3040/60 X'per PRO, PANalytical, using Cu-Kα (λ = 1.54056 Å) radiation, an accelerating voltage of 40 kV and a current of 30 mA. Ultraviolet–visible absorbance was acquired on a UV–Vis spectrometer (UV-2401PC, Shimadzu). Synchrotron PXRD measurements (X-ray energy = 18.986 keV, λ = 0.65303 Å) were performed at 6D beamline of the Pohang Accelerator Laboratory. X-ray photoelectron spectroscopy (XPS) and XPS depth profiling were carried out on a Thermo-Fisher machine (ESCALAB 250XI) using Al K$_\alpha$ source. Time-of-flight secondary ion mass spectrometry (TOF-SIMS) was carried out on a TOF-SIMS V instrument (ION-TOF GmbH, Germany) using a 10 keV Bi$^+$ primary ion beam source. The etched depths were measured with a surface profiler (KLA Tencor P6 Profilometer). The morphology of the samples was observed by a field-emission scanning electron microscope (FESEM-S4800, HITACHI). High-angle annular dark-field scanning transmission electron microscopy (HAADF-STEM) images and corresponding electron energy loss spectroscopy (EELS) mapping were taken using a FEI Titan3 G2 60-300 microscope equipped with a double-sided Cs corrector operating at 200 kV.

**(Photo)electrochemical measurements.** All (photo)electrochemical measurements were conducted on a potentiostat (IviumStat, Ivium Technologies) in 1 M KOH electrolyte under the 1-sun condition (100 mW cm$^{-2}$) by a solar simulator (91160, Oriel) equipped with an air mass (AM) 1.5 G filter. Note that the exposed surface area of all the prepared photoanodes was geometric area (1 × 1 cm, other parts were covered with epoxy resin) and the electrolyte was slowly stirred during the measurements. All the measured potentials vs. Ag/AgCl reference electrode were converted to the potentials vs. reversible hydrogen electrode (RHE) by the Nernst equation: $E_{RHE} = E_{Ag/AgCl} + 0.059 \text{ pH} + E^o_{Ag/AgCl}$ ($E^o_{Ag/AgCl} = 0.1976$ at 25 °C). The potentials were swept from 0.4 to 1.8 $V_{RHE}$ at a scanning rate of 20 mV s$^{-1}$. The open circuit potential (OCP) transient decay was measured in the same experimental condition as PEC measurements. On the open circuit condition, the open circuit potential of working electrode was first stabilized for 10 min under illumination, and then the light source was turned off to record the OCP decay continuously for the next 10 min.

The photoelectrochemical impedance spectroscopy (PEIS) were obtained from 0.7 to 1.3 $V_{RHE}$ under simulated 1-sun condition with a frequency range 0.1 Hz—100 kHz. Nyquist plots (imaginary vs. real components of impedance, $Z_{img}$ vs. $Z_{real}$) were fitted to the corresponding equivalent circuits by Z-view software. Mott–Schottky plots were obtained by sweeping in the range of 0.1−1.0 $V_{RHE}$ with AC frequency of 1000 Hz under the dark condition. The incident photon-to-current conversion efficiency (IPCE) was estimated by a Xe lamp (300 W, Oriel) and a monochromator with a band width of 5 nm at 1.23 $V_{RHE}$ in the same electrolyte (1 M KOH).

$$\text{IPCE} = \frac{1240 \times J(\text{mA cm}^{-2})}{P_{light}(\text{mW cm}^{-2}) \times \lambda(\text{nm})} \qquad (3)$$

where $J$ is the measured photocurrent density using a monochromator, $P_{light}$ is the calibrated illumination power at the specific wavelength, and $\lambda$ is the corresponding wavelength of the incident light. The charge–carrier density, $N_D$, is inversely

proportional to the slope of Mott–Schottky plot:

$$\left(\frac{A_S}{C_{bulk}}\right)^2 = \frac{2}{q\varepsilon_r\varepsilon_0 N_D}\left(V - E_{FB} - \frac{k_B T}{q}\right) \quad (4)$$

with $\left(\frac{A_S}{C_{bulk}}\right)$ being the surface area-corrected space-charge capacitance, $V$ the applied potential, $E_{FB}$ the flat-band potential of the electrode, $k_B = 1.38 \times 10^{-23}$ J K$^{-1}$, $T = 298$ K, $q = 1.602 \times 10^{-19}$ C, $\varepsilon_0 = 8.85 \times 10^{-12}$ C$^2$ J$^{-1}$;m$^{-1}$, and $\varepsilon_r = 32$ for hematite. The amounts of H$_2$ and O$_2$ gases evolved from a closed circulation PEC system were analyzed by a gas chromatograph (GC, HP 7890) equipped with a thermal conductivity detector (TCD) and a packed column (Supelco, Carboxen 1000) using high purity Ar as a carrier gas (at a constant rate of 10 mL min$^{-1}$).

**Transient absorption spectroscopy (TAS)**. TAS measurements were carried out using the third harmonic of a Nd:YAG laser (EKSPLA, NT 342B, 355 nm, 5 ns pulse width, 0.9 Hz) as the excitation source. A liquid light guide transmitted the laser pulse to the sample resulting in an incident pump intensity of ca. 177 μJ cm$^{-2}$ (355 nm). A 100 W tungsten lamp (Bentham, IL 1) coupled to a monochromator (Zolix, Omni - λ 300) was used as the probe light. Variation in optical density (ΔOD) of the sample was calculated by measuring the transmitted light using a Si photodiode (Hamamatsu) and an amplification system coupled to both an oscilloscope (Tektronix, TDS 2012C) and data acquisition card (National Instruments NI-6221). The data were averaged over 400 laser shots. OCP measurements were conducted in 1 M NaOH solution with or without 0.5 M Na$_2$SO$_3$. The *operando* TAS experiments were implemented by three-electrode setup controlled by a CHI 760C potentiostat in 1M NaOH solution (pH = 13.7), with the photoanode, Pt, and Ag/AgCl as working, counter and reference electrodes, respectively. During TAS measurements, a constant potential was maintained by chronoamperometry.

## Data availability

The data that support the plots within this paper and other findings of this study are available from the corresponding author upon reasonable request. The source data underlying Figs. 2g, 3, 4, 5 and 6 and Supplementary Figs. 2, 3a, 6, 7, 8, 10, 11, 13, 14, 15d–f, 16, 17, 19, 20, 22, 23, 24, 25, 27, 29, 30, 31, 32, 33, 34, 35 and 36 are provided as a Source Data file. Source data are provided with this paper.

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

# ARTICLE

42. Cao, D. et al. Cathodic shift of onset potential for water oxidation on a $Ti^{4+}$ doped $Fe_2O_3$ photoanode by suppressing the back reaction. *Energy Environ. Sci.* **7**, 752–759 (2014).

43. Zaban, A., Greenshtein, M. & Bisquert, J. Determination of the electron lifetime in nanocrystalline dye solar cells by open-circuit voltage decay measurements. *ChemPhysChem* **4**, 859–864 (2003).

44. Zhong, M. et al. Surface Modification of $CoO_x$ Loaded $BiVO_4$ Photoanodes with Ultrathin p-Type NiO Layers for Improved Solar Water Oxidation. *J. Am. Chem. Soc.* **137**, 5053–5060 (2015).

45. John A. D. *Lange's handbook of chemistry* (McGraw-Hill, 1998).

46. Klahr, B., Gimenez, S., Fabregat-Santiago, F., Hamann, T. & Bisquert, J. Water oxidation at hematite photoelectrodes: the role of surface states. *J. Am. Chem. Soc.* **134**, 4294–4302 (2012).

47. Qiu, J. et al. Direct in situ measurement of charge transfer processes during photoelectrochemical water oxidation on catalyzed hematite. *ACS Cent. Sci.* **3**, 1015–1025 (2017).

48. Zhang, J., García-Rodríguez, R., Cameron, P. & Eslava, S. Role of cobalt–iron (oxy)hydroxide (CoFeO*x*) as oxygen evolution catalyst on hematite photoanodes. *Energy Environ. Sci.* **11**, 2972–2984 (2018).

49. Monllor-Satoca, D. et al. What do you do, titanium? Insight into the role of titanium oxide as a water oxidation promoter in hematite-based photoanodes. *Energy Environ. Sci.* **8**, 3242–3254 (2015).

50. Sivula, K. Metal oxide photoelectrodes for solar fuel production, surface traps, and catalysis. *J. Phys. Chem. Lett.* **4**, 1624–1633 (2013).

51. Pesci, F. M., Wang, G., Klug, D. R., Li, Y. & Cowan, A. J. Efficient suppression of electron–hole recombination in oxygen-deficient hydrogen-treated $TiO_2$ nanowires for photoelectrochemical water splitting. *J. Phys. Chem. C.* **117**, 25837–25844 (2013).

52. Barroso, M. et al. Dynamics of photogenerated holes in surface modified α-$Fe_2O_3$ photoanodes for solar water splitting. *Proc. Natl Acad. Sci. USA* **109**, 15640–15645 (2012).

53. Barroso, M., Pendlebury, S. R., Cowan, A. J. & Durrant, J. R. Charge carrier trapping, recombination and transfer in hematite (α-$Fe_2O_3$) water splitting photoanodes. *Chem. Sci.* **4**, 2724–2734 (2013).

54. Forster, M., Potter, R. J., Yang, Y., Li, Y. & Cowan, A. J. Stable $Ta_2O_5$ overlayers on hematite for enhanced photoelectrochemical water splitting efficiencies. *ChemPhotoChem* **2**, 183–189 (2018).

55. Sachs, M. et al. Effect of oxygen deficiency on the excited state kinetics of $WO_3$ and implications for photocatalysis. *Chem. Sci.* **10**, 5667–5677 (2019).

56. Francàs, L. et al. Spectroelectrochemical study of water oxidation on nickel and iron oxyhydroxide electrocatalysts. *Nat. Commun.* **10**, 5208 (2019).

57. Zandi, O. & Hamann, T. W. Enhanced water splitting efficiency through selective surface state removal. *J. Phys. Chem. Lett.* **5**, 1522–1526 (2014).

## Acknowledgements
This work was supported by the Climate Change Response project (2019M1A2A2065612), the Basic Science Grant (NRF-2019R1A4A1029237), the National Natural Science Foundation of China (No. 21872143), and Korea-China Key Joint Research Program (2017K2A9A2A11070341) funded by MSIT, Pohang University of Science and Technology (POSTECH), and UCRF at UNIST.

## Author contributions
H.Z. conceived the study, organized the collaboration, conducted the experiments, analyzed the data, and wrote the manuscript; D.L. and X.W. carried out TAS measurements and analyzed the results; H.Y.J. performed STEM measurements; W.J.B. assisted Faraday efficiency measurements; T.J.S. performed synchrotron PXRD measurements, H.H. and C.L. supervised the collaboration project in China; and J.S.L. supervised the collaboration project in South Korea, and contributed to the writing of the manuscript. All authors discussed the results and commented on the manuscript.

## Competing interests
The authors declare no competing interests.
