## [Peer Review File · Nature Communications]

Reviewers' comments:

Reviewer #1 (Remarks to the Author):

The manuscript entitled "Gradient tantalum-doped hematite homojunction photoanode improves both photocurrents and turn-on voltage for solar water splitting" described a novel nanostructured hematite-based photoanode wherein the synergistic combination of surface and bulk engineering results in a high-performance photoanode for water oxidation. The article characterizes in great detail the new photoanode, both the structure and composition and also the PEC response to decipher the reason for the improved response with respect a conventional nanostructured Fe₂O₃ electrode. Although the study is interesting and certainly it has been done with great care, there exist some key issues that lead me to do not recommend for publication as it is. But, I would encourage the authors to work on some of these issues prior next submission.

1) The authors emphasize along the text that the performance they obtained is among the best reported page 11, line 206 "This performance is superior or comparble ((typo : comparable) to the recently reported state-of-the-art...". But the authors forgot to mention Choi's work (Nano Energy 2017, 39, 211-218) where the authors reported 6 mA cm⁻² @ 1.23 V vs RHE, and a fairly low onset potential at about 0.58 V vs RHE. I would suggest the authors to mention this work as well and highlight some the advantages of their approach.

2) One of probably the weakest parts of the paper is the attempt to unambiguously clarify the reasons for the enhancement of the performance. The authors employ PEIS, charge separation/injection efficiency measurements and TAS. However, there are several aspects that need to be clarify. For instance, the NiFe(OH) used by the authors has been reported to work as either electrocatalyst in some cases but also as passivating agent (Chem. Sci., 2016, 7, 3347 ; ACS Energy Lett. 2018, 3, 4, 961-969). The authors in the text, described it as an electrocatalysts but sometimes they invoke that it is passivating electron traps. The authors should be more clear on the beneficial role of this layer. One of the possibilities could be to employ OCP measurements or IMPS as Wang's (Nat Commun 6, 7447 (2015)) group has reported before to clarify the actual role of the this overlayer.

3) The authors include some measurements performed by using TAS, a very powerful technique that could certainly clarify how the treatments (Ta doping; FeOOH regrowth and NiFe(OH)) improve the response. The authors probe two different wavelengths to either track the trapped electrons or the photogenerated holes. Unfortunately, the measurements under OCP do not provide any significant insights, e.g. they observed a large electron trapping that correlates well with the transient spikes in the photocurrent, or a larger amount of holes in the Ta:Fe₂O₃@Fe₂O₃. But some data is difficult to grasp, for instance, why in the presence of NiFeOH the amount of photoholes are lower? Are they transferred to the NiFeOOH? If so, could the authors monitor the signal of the holes within the NiFeOOH? I would suggest the authors to include data recorded in the presence of electrolyte (for instance NaOH or Na₂SO₃) to better understand the carrier dynamics and how the subsequent treatments affect the performance. Also, by including some measurement with applied bias could shed some light into how the trapping phenomena changes with the treatments.

Overall, although the characterization of the structure and crystallinity of the material is very comprehensive, the fact that the performance is not a benchmark in the field requires bigger efforts on the PEC characterization to really differentiate this work and provide unique insights on what is the role of each treatment and how it improves the response. For all these reasons, given the high standards for publication in nature communication I am not able to accept for publication as it is, but dedicating some time to improve the PEC response could certainly improve the novelty of the paper.

Reviewer #2 (Remarks to the Author):

The present manuscript reports on gradient tantalum doped hematite photo-anodes modified with a NiFe(OH)_x co-catalyst (NiFe(OH)_x/Ta-Fe₂O₃@Fe₂O₃) showing optimized photocurrent density of 3.22

mA cm⁻² at 1.23 V RHE and onset potential of 0.55 V RHE under simulated solar light illumination. The hematite nanorods were formed by hybrid microwave annealing (HMA), a method introduced by the authors in previous work.

In my view, in spite of the significantly enhanced PEC performance of NiFe(OH)_x/Ta-Fe₂O₃@Fe₂O₃, the present manuscript requires revision, i.e. more detailed information, according to the points below for publication in Nature Communication:

- There are various published papers about gradient doping (e.g. Chem. Sci., 2017, 8, 91-100; ChemSusChem, 2018, 11, 1873-1879; J. POWER SOURCES, 2020, 449, 227473; Phys. Chem. Chem. Phys., 2016, 18, 32735-32743). The effects of gradient doping described in the present manuscript are mainly an increased J_{ph} and a cathodic shift of V_{on} which are very similar to what already reported. However, the authors find for Ta-Fe₂O₃@Fe₂O₃ a much higher performance than previous studies. Hence, the reviewer wonders what makes the gradient Ta ion doping here presented unique compared with other doping strategies (Ti, Sn, Pt or P ion doping) previously reported. It is somehow hard to accept such a significantly enhanced PEC performance based on the present plain explanation.
- The authors should add the etching rate used for the ToF-SIMS measurements and for the depth profiling XPS analysis; this information is very important to verify the gradient doping (also sputter artifacts should be excluded).
- The authors should add the results of electrochemical measurements of pristine hematite as a reference. Particularly important is the comparison of Mott-Schottky data of NiFe(OH)_x/Ta-Fe₂O₃@Fe₂O₃ vs. pristine hematite.
- We strongly suggest the authors to apply a mask in front of the photo-anodes for the PEC measurements, this to define the activate area in hematite and to prevent any influences from light scattering which may enhance the PEC performance. Furthermore, we would kindly ask the authors to provide a picture of their PEC measurement system.

Overall, I recommend this manuscript to be strongly revised before considering publication in Nature Communications.

Reviewer #3 (Remarks to the Author):

This work reported a core-shell formation of gradient tantalum-doped hematite homojunction nanorods that improves photocurrents and turn-on voltage for photoelectrochemical water splitting. The authors claimed that homojunction formation could passivate the surface states and provide an electric field to suppress charge recombination, leading to the enhanced photocurrents and turn-on voltage. Besides, their homojunction presented a synergy with NiFe(OH)_x cocatalysts with additional improvements of photocurrents and turn-on voltage. This work is novel and provides some insights for designing efficient semiconductor photoelectrodes for photoelectrochemical water splitting, which could be recommended for a publication in Nature Communications after addressing the following comments.

1. What is the difference between tantalum-doped hematite vs. bare hematite? Provide photoelectrochemical performance and other data of Fe₂O₃ and Fe₂O₃@Fe₂O₃.
2. What are the states of Ta atoms in the Ta:Fe₂O₃ and Ta:Fe₂O₃@ Fe₂O₃, are they isolated atoms or nanoclusters? Only HRSTEM data is not sufficient.
3. There is only one STEM image of Ta:Fe₂O₃ and Ta:Fe₂O₃@ Fe₂O₃ in the paper, provide more STEM data on other Ta:Fe₂O₃ and Ta:Fe₂O₃@ Fe₂O₃ to see the reproducibility.
4. Page 10, Figure 3a, why are there no Ta peaks in the XRD patterns?
5. There are no physical characterizations of NiFe(OH)_x/Ta:Fe₂O₃ and NiFe(OH)_x/Ta:Fe₂O₃@Fe₂O₃,

perform these measurements.

6. Provide more details about the information of the photoelectrochemical cell.

7. Page 14, provide more details about how the H₂ and O₂ were measured. Run multiple sets of gas analysis experiments to get statistics.

Response to Editor and Reviewers' Comments

We like to thank the editor and the reviewers for spending their time to carefully evaluate this work and make valuable comments and suggestions, which were of great help in improving the quality of this work. All the comments have been accommodated as summarized in the following point-by-point responses. The revised parts of the manuscript have been highlighted.

Reviewer #1

Comments to the Author: The manuscript entitled “Gradient tantalum-doped hematite homojunction photoanode improves both photocurrents and turn-on voltage for solar water splitting” described a novel nanostructured hematite-based photoanode wherein the synergistic combination of surface and bulk engineering results in a high-performance photoanode for water oxidation. The article characterizes in great detail the new photoanode, both the structure and composition and also the PEC response to decipher the reason for the improved response with respect a conventional nanostructured Fe_2O_3 electrode. Although the study is interesting and certainly it has been done with great care, there exist some key issues that lead me to do not recommend for publication as it is. But, I would encourage the authors to work on some of these issues prior next submission.

Comment 1): The authors emphasize along the text that the performance they obtained is among the best reported page 11, line 206 “This performance is superior or comparble ((typo: comparable) to the recently reported state-of-the-art...”. But the authors forgot to mention Choi’s work (Nano Energy 2017, 39, 211-218) where the authors reported 6 mA cm^{-2} @ 1.23 V vs RHE, and a fairly low onset potential at about 0.58 V vs RHE. I would suggest the authors to mention this work as well and highlight some the advantages of their approach.

Response 1): As the reviewer suggested, Choi’s work has been cited and described in the revised manuscript as follows together with the correction of the typo of “comparable”.

(Line 13, Page 3)

In particular, hematite ($\alpha\text{-Fe}_2\text{O}_3$) is an ideal candidate material in PEC water splitting due to its small band gap (2.1 eV) that allows a high theoretical STH efficiency (15.5 %) in addition to the mentioned typical advantages of metal oxides.⁷ Notably, a benchmark record of $\sim 6 \text{ mA cm}^{-2}$ at 1.23 V_{RHE} under standard conditions was reported by Jeon *et al.*⁸ using combined modifications of hydrogen treatment, ultrathin TiO_2 overlayer, and cobalt phosphate cocatalyst loading. Even this superb performance corresponds to less than 50 % of its theoretical limit.

We also added their results in Supplementary Table 2 for comparison with other works and cited this work as Ref. 8.

Ref:

8. Jeon, T. H., Moon, G.-h., Park, H. & Choi, W. Ultra-efficient and durable photoelectrochemical water oxidation using elaborately designed hematite nanorod arrays. *Nano Energy* **39**, 211-218 (2017).

Comment 2): One of probably the weakest parts of the paper is the attempt to unambiguously clarify the reasons for the enhancement of the performance. The authors employ PEIS, charge separation/injection efficiency measurements and TAS. However, there are several aspects that need to be clarify. For instance, the NiFe(OH)_x used by the authors has been reported to work as either electrocatalyst in some cases but also as passivating agent (Chem. Sci., 2016, 7, 3347 ; ACS Energy Lett. 2018, 3, 4, 961-969). The authors in the text, described it as an electrocatalysts but sometimes they invoke that it is passivating electron traps. The authors should be more clear on the beneficial role of this layer. One of the possibilities could be to employ OCP measurements or IMPS as Wang's (Nat. Commun. 6, 7447 (2015)) group has reported before to clarify the actual role of this overlayer.

Response 2): We thank the reviewer's advice to employ OCP measurements or IMPS in order to clarify the actual role of NiFe(OH)_x overlayer. The NiFe(OH)_x overlayer, as various other co-catalyst overlayers (such as "CoPi", FeOOH, NiOOH, and NiFeO_x), has the similar dual effects to passivate surface trapping states of the photoelectrode, in addition to the role of a conventional "catalyst" that accelerates water oxidation reaction as well studied experimentally and theoretically (*J. Mater. Chem. A* **2015**, 3, 20649;

Chem. Sci. **2016**, 7, 3347; *ACS Energy Lett.* **2016**, 1, 624; *Chem. Mater.* **2017**, 29, 6674; *ACS Catal.* **2018**, 8, 2754). The catalytic role of the metal oxide/(oxy)hydroxide overlayers involves hole collectors and oxygen-evolution catalysis to promote the rather sluggish photoanode reaction. (*J. Am. Chem. Soc.* **2019**, 141, 1394; *ACS Cent. Sci.* **2017**, 3, 1015; *ACS Energy Lett.* **2018**, 3, 961; *Nat. Energy* **2018**, 3, 46). Our results including PEIS, OCP decay and TAS suggest that **the beneficial role of NiFe(OH)_x cocatalyst layer on hematite photoanodes should include both passivation of surface states and oxygen-evolution catalysis**, which is consistent with common understanding of the community on the roles of the catalyst overlayers on metal oxide based photoanodes.

As the reviewer suggested, OCP measurements have been added in the revised manuscript (Fig. 4i and Supplementary Fig. 24). Although OCP cannot directly clarify the actual role of NiFe(OH)_x layer, it does help us to see the existence of built-in electric field and how it affects the behavior of the charges. Following statements have been added or modified.

(Line 6, Page 16)

The open circuit potential (OCP) transient decay profile can provide additional information on built-in electric field and its effects on the behaviour of generated charges. Compared with Ta:Fe₂O₃, the homojunction photoanode shows a remarkably accelerated OCP decay, indicative of a larger photovoltage generation ($\Delta OCP = OCP_{\text{dark}} - OCP_{\text{light}}$) (Supplementary Fig. 24), which demonstrates that homojunction can provide an additional intrinsic built-in electric field to enhance the driving force of charge separation. The OCP decay is further enhanced by loading NiFe(OH)_x cocatalyst on the homojunction, confirming the effectiveness of cocatalyst modification. For comparison of charge recombination rate at the photoanode/electrolyte junction, the carrier lifetime was quantified by:⁴³

$$\tau_n = -\frac{\kappa_B T}{e} \left(\frac{dOCP}{dt} \right)^{-1}$$

(1)

where τ_n , k_B , T , e and $dOCP/dt$ are the carrier lifetime, Boltzmann's constant, temperature (K), charge of single electron and derivative of the OCP transient decay, respectively. The carrier lifetime of homojunction is ~ 190 ms at the transient when illumination is stopped (Fig. 4i), which is smaller by a factor of 23 compared to ~ 4.4 s for Ta:Fe₂O₃. The fast decay kinetics is indicative of the enhanced charge recombination when illumination is removed, which suggests on the other hand that that charge trapping is insignificant when illuminated.^{44, 45} Consequently, the effective charge separation is expected in the homojunction photoanode under PEC water splitting condition.

(Line 19, Page 26)

The open circuit potential (OCP) transient decay was measured in the same experimental condition as PEC measurements. On the open circuit condition, the open circuit potential of working electrode was first stabilized for 10 min under illumination, and then the light source was turned off to record the OCP decay continuously for the next 10 min.

Fig. 4i OCP-derived carrier lifetime for Ta:Fe₂O₃, Ta:Fe₂O₃@Fe₂O₃, and NiFe(OH)_x/Ta:Fe₂O₃@Fe₂O₃ photoanodes.

Supplementary Fig. 24| OCP transient decay profiles for Ta:Fe₂O₃, Ta:Fe₂O₃@Fe₂O₃, NiFe(OH)_x/Ta:Fe₂O₃@Fe₂O₃. The OCP transient decay evaluates the surface recombination between trapped electrons and reaction intermediates rather than the bulk recombination and this process is very fast (within ns–ms domain), while the scale of OCP decay is usually in several minutes.¹⁰ Generally, OCP is very positive in dark due to a largely upward band bending, while OCP will be more cathodic under illumination due to flattening of the energy band by photoexcited carriers. Δ OCP ($OCP_{\text{dark}} - OCP_{\text{light}}$), also known as photovoltage, represents the amount of band bending under illumination with respect to that in the dark condition. Homojunction formation boosts Δ OCP value compared to that of Ta:Fe₂O₃, indicating that homojunction can generate an additional built-in electric field. At the transient time from illumination (quasi-equilibrium of flattened energy band) to the dark (equilibrium of bent energy band), the charge recombination strongly depends on the spatial charges built in the photoanode/electrolyte junction. The strong band bending enables a large amount of spatial charges in the depletion region and thus significant charge recombination occurs at the transient when the illumination is removed. As such, a fast OCP-decay is expected.

Refs:

43. Zaban, A., Greenshtein, M. & Bisquert, J. Determination of the electron lifetime in nanocrystalline dye solar cells by open-circuit voltage decay measurements. *ChemPhysChem* **4**, 859-864 (2003).

44. Lin, Y., et al. Growth of p-type hematite by atomic layer deposition and its utilization for improved solar water splitting. *J. Am. Chem. Soc.* **134**, 5508-5511 (2012).
45. Zhong, M., et al. Surface Modification of CoO_x Loaded BiVO_4 Photoanodes with Ultrathin p-Type NiO Layers for Improved Solar Water Oxidation. *J. Am. Chem. Soc.* **137**, 5053-5060 (2015).
10. Yi S-S, Wulan B-R, Yan J-M, Jiang Q. Highly efficient photoelectrochemical water splitting: surface modification of cobalt-phosphate-loaded $\text{Co}_3\text{O}_4/\text{Fe}_2\text{O}_3$ p-n heterojunction nanorod arrays. *Adv Funct Mater* **29**, 1801902 (2019).

Comment 3): *The authors include some measurements performed by using TAS, a very powerful technique that could certainly clarify how the treatments (Ta doping; FeOOH regrowth and $\text{NiFe}(\text{OH})_x$) improve the response. The authors probe two different wavelengths to either track the trapped electrons or the photogenerated holes. Unfortunately, the measurements under OCP do not provide any significant insights, e.g. they observed a large electron trapping that correlates well with the transient spikes in the photocurrent, or a larger amount of holes in the $\text{Ta}:\text{Fe}_2\text{O}_3@\text{Fe}_2\text{O}_3$. But some data is difficult to grasp, for instance, why in the presence of $\text{NiFe}(\text{OH})_x$ the amount of photoholes are lower? Are they transferred to the NiFeOOH ? If so, could the authors monitor the signal of the holes within the NiFeOOH ? I would suggest the authors to include data recorded in the presence of electrolyte (for instance NaOH or Na_2SO_3) to better understand the carrier dynamics and how the subsequent treatments affect the performance. Also, by including some measurement with applied bias could shed some light into how the trapping phenomena changes with the treatments.*

Response 3): We thank the reviewer for this comment. As suggested, we investigated charge dynamics of Fe_2O_3 photoanodes in various electrolytes and with applied bias through TAS. During TAS retests in 1 M NaOH solution, we found that the PEC measurements with external bias will greatly influence subsequent TAS dynamics at OCP, as shown in Figure R1. The absorption and bleach amplitudes under OCP displayed an obvious increase when the PEC measurements were closely followed with the external bias. Moreover, the TAS decay dynamics showed the different decay processes relative to that of OCP without influence by the applied bias. In order to eliminate the interference of photoelectrocatalysis or bias, the TAS dynamic measurements at OCP were implemented again by holding the samples in the dark without bias for more than 24

hours. For the accuracy, the TAS data under OCP in Fig. 6a, b have been reedited in the revised manuscript.

Fig. R1 TAS of $\text{NiFe(OH)}_x/\text{Ta:Fe}_2\text{O}_3@\text{Fe}_2\text{O}_3$ under OCP condition by following PEC measurements (denoted as After PEC) and by holding the sample in the dark without bias for more than 24 hours (denoted as Before PEC).

The TAS results under OCP condition reflect the photogenerated charge recombination process without the influence of reaction processes. Thus the effect of Ta doping, homojunction and NiFe(OH)_x on charge recombination can be revealed directly. TAS data demonstrate that Ta doping increases trap states dramatically, as revealed by the amplitude of the bleach at 580 nm of $\text{Ta:Fe}_2\text{O}_3$ while there is almost no bleach signal in pristine Fe_2O_3 . In contrast, the formation of homojunction greatly suppresses charge-carrier recombination and generates a larger amount of residual photoholes in comparison with that of $\text{Ta:Fe}_2\text{O}_3$. Subsequent modification with NiFe(OH)_x passivates the surface electron trap states effectively, which is illustrated by the dramatically decreased amplitude of bleaching signal at 580 nm relative to that of homojunction sample. For the reduced TAS intensity at OCP after NiFe(OH)_x modification, this reduction is assigned to the passivation of electron trapped states. Ideally, all the photoinduced charge carriers in hematite will recombine eventually under OCP condition. The existence of trapped states will trap the photoinduced electrons and elongate its recombination process, leading to

the decrease of direct recombination. Consequently, the passivation of electron trapped states will enhance direct recombination path resulting in decreased TAS intensity. From a previous report of NiFeOOH as electrocatalyst (*Nat. Commun.* **2019**, 10, 5208), the hole absorption spectra in NiFeOOH is overlapped with photohole in Fe₂O₃. In addition, it was suggested that the reactive center in NiFeOOH electrocatalyst is Fe-site, but it cannot be distinguished from the hole in hematite (*Nat. Energy* **2020**, 5, 222). In fact, the TAS decay shapes for two photoanodes with/without NiFe(OH)_x under 1.3 V_{RHE} are overlapped. Thus, it is impossible to directly monitor/distinguish hole signal within NiFe(OH)_x in the present system.

For the question in different electrolytes (for instance NaOH or Na₂SO₃), the TAS measurements were done in air, NaOH or Na₂SO₃ to better understand the carrier dynamics and how the subsequent treatments affect the performance (Supplementary Fig. 32, 33, 34). In comparison with TAS results in air, TAS results in NaOH or Na₂SO₃ electrolytes show that the formation of semiconductor/liquid junction actually affected the recombination processes. Using Na₂SO₃ as a hole scavenger, the transient signal at 580 and 694.3 nm were assigned to photoelectrons and photoholes, respectively. Moreover, combined with the previous report that hematite is insensitive in sacrificial reagent (*Chem. Commun.* **2011**, 47, 716), the photoinduced charges in Ta-doped photoanodes can migrate to photoelectrode surface more easily.

For the suggestion that some measurements with applied bias could shed some light into how the trapping phenomena changes with the treatments, we have carried out bias-dependent TAS measurements (Fig. 6c, d and Supplementary Fig. 36). From the TAS results of 580 nm at bias, Ta doping prolongs the detrapped time of the trapped photoelectrons. Homojunction formation and further NiFe(OH)_x modification accelerate this process and decrease the detrapped time. Moreover, the TAS dynamics at 694.3 nm unravel that homojunction and NiFe(OH)_x modification can efficiently improve the utilization of photoholes from the view of timescale. According to the reviewer's suggestion of TAS, we have added the TAS data of the three photoanodes in various electrolytes as well as with applied bias. Revision was made in highlight as follows:

(Second last line, Page 21)

Hole scavenger reagent of Na_2SO_3 was chosen to confirm the TAS signal (Supplementary Fig. 32). As expected, Na_2SO_3 consumes photoholes evidenced by decreased amplitude of 694.3 nm absorption, resulting in more photoelectrons revealed by the increased amplitude of 580 nm.

Under OCP condition where the electrodes were immersed in 1 M NaOH electrolyte without external bias (Fig. 6a, b and Supplementary Fig. 33a), the photogenerated charge-carrier dynamics in hematite photoanodes should be assigned to its bulk electron-hole recombination processes, since their decay dynamics are faster than the timescale of water oxidation on hematite surface as shown in bias-dependent TAS dynamics (Fig. 6d). Nevertheless, the existence of semiconductor/liquid junction will affect the recombination processes of charge/carrier relative to that in air (Supplementary Fig. 34). Compared with Ta: Fe_2O_3 photoanode, TAS of homojunction photoanode shows a slightly reduced bleach amplitude (Fig. 6a), indicating some passivation of electron trap states.⁵⁵ More importantly, the homojunction photoanode shows a remarkable retardation of the fast decay curves and results in a long half-lifetime ($\tau_{50\%} = 1.4$ ms) at the 694.3 nm transient relative to Ta: Fe_2O_3 ($\tau_{50\%} = 0.23$ ms), demonstrating that homojunction greatly suppresses charge-carrier recombination and generates a larger amount of residual photoholes. These results further confirm the existence of built-in electric field between the core (Ta: Fe_2O_3) and the shell (Fe_2O_3) parts of homojunction with the field direction from inside to outside, which significantly promotes photogenerated charge separation and accumulation. For the $\text{NiFe}(\text{OH})_x$ modified homojunction, the bleach amplitude at 580 nm decreases dramatically relative to the bare homojunction photoanode, which suggests the large decrease of trapped photoelectrons by passivation of surface electron trap states.⁵⁵ On the other hand, the absorption amplitude at 694.3 nm also slightly decreases and the half-lifetime decreases to 0.19 ms due to the reduction of electron trap states, resulting from the increase of direct recombination⁵⁶ or the charge transfer from homojunction to $\text{NiFe}(\text{OH})_x$.⁵⁷

To understand the actual PEC processes, the *operando* TAS experiments were implemented under applied bias of 1.3 V_{RHE} (Fig. 6c, d and Supplementary Fig. 33b). The bleach recovery of Ta:Fe₂O₃@Fe₂O₃ ($\tau_{50\%} = 0.90$ ms) and Ta:Fe₂O₃@Fe₂O₃/NiFe(OH)_x ($\tau_{50\%} = 0.51$ ms) photoanodes is much faster than that of Ta:Fe₂O₃ ($\tau_{50\%} = 1.3$ ms), which illustrates that the trapped electrons were extracted more efficiently. In terms of the absorption decay dynamics at 694.3 nm (1.3 V_{RHE}), three photoanodes exhibit similar biphasic decay including electron-hole recombination and water oxidation: a fast decay phase (< 10 ms) assigned to electron-hole recombination at early times, and a slow decay phase (10 ms ~ s) assigned to the population of long-lived photoholes for water oxidation. Ta:Fe₂O₃@Fe₂O₃ (5.5 ms) shows a much shorter time relative to that of Ta:Fe₂O₃ (11 ms) at the turning point of the fast and slow phases (Supplementary Fig. 35), indicating that homojunction formation can efficiently employ photoholes at space charge region from the view of timescale. In addition, bias-dependent TAS results in Supplementary Fig. 36 display that the long-lived holes appear at lower bias (0.7–0.8 V_{RHE}) in homojunction photoanode than that in Ta:Fe₂O₃ photoanode (0.8–0.9 V_{RHE}), which agrees well with the negative shift of onset potential. Therefore, the homojunction formation with Fe₂O₃ surface is beneficial to take advantage of photoholes at the space charge region (Supplementary Fig. 37). Moreover, NiFe(OH)_x modification can further shorten the time to achieve turning point (2.3 ms), revealing the promotion effect in surface reaction dynamics. By mono-exponential fitting for timescale of 10 ms ~ 1 s, homojunction samples with/without NiFe(OH)_x modification display more long-lived holes, which will facilitate the PEC performance.

(Line 6, Page 29)

OCP measurements were conducted in 1 M NaOH solution with or without 0.5 M Na₂SO₃. The *operando* TAS experiments were implemented by three-electrode setup controlled by a CHI 760C potentiostat in 1M NaOH solution (pH = 13.7), with the photoanode, Pt, and Ag/AgCl as working, counter and reference electrodes, respectively. During TAS measurements, a constant potential was maintained by chronoamperometry.

Fig. 6 Transient absorption spectroscopy (TAS) for Ta:Fe₂O₃, Ta:Fe₂O₃@Fe₂O₃, and NiFe(OH)_x/Ta:Fe₂O₃@Fe₂O₃ photoanodes. (a) OCP at 580 nm. (b) OCP at 694.3 nm. (c) 1.3 V_{RHE} at 580 nm. (d) 1.3 V_{RHE} at 694.3 nm.

Supplementary Fig. 32 | Transient absorption spectroscopy (TAS) for Ta:Fe₂O₃, Ta:Fe₂O₃@Fe₂O₃, and NiFe(OH)_x/Ta:Fe₂O₃@Fe₂O₃ photoanodes in electrolyte of 1 M NaOH with or without 0.5 M Na₂SO₃.

Supplementary Fig. 33 | TAS and corresponding half lifetime ($\tau_{50\%}$) for Ta:Fe₂O₃, Ta:Fe₂O₃@Fe₂O₃ and NiFe(OH)_x/Ta:Fe₂O₃@Fe₂O₃ photoanodes. **(a)** OCP at 694.3 nm. **(b)** 1.3 V_{RHE} at 580 nm. $\tau_{50\%}$ is the time when the TAS amplitude decreases by half of its initial value at 10 μ s.¹¹

Supplementary Fig. 34 | TAS for $Ta:Fe_2O_3$, $Ta:Fe_2O_3@Fe_2O_3$, and $NiFe(OH)_x/Ta:Fe_2O_3@Fe_2O_3$ photoanodes in air. In air, three samples show negligible absorption at 694.3 nm and fast decay at 580 nm. The order of amplitude at 10 μ s is as follows: $Ta:Fe_2O_3@Fe_2O_3 > NiFe(OH)_x/Ta:Fe_2O_3@Fe_2O_3 > Ta:Fe_2O_3$. It demonstrates that Ta doping and homojunction fabrication can promote the formation of long-lived photoinduced charges.

Supplementary Fig. 35 | The turning point of biphasic decay at 694.3 nm and 1.3 V_{RHE} for Ta:Fe₂O₃ (a), Ta:Fe₂O₃@Fe₂O₃ (b) and NiFe(OH)_x/Ta:Fe₂O₃@Fe₂O₃ (c) photoanodes. At 1.3 V_{RHE}, the decay dynamics at 694.3 nm displays biphasic decay, including recombination process at μ s – ms and water oxidation process at ms – s.¹² To compare the timescale of starting water oxidation, the turning point is extracted from biphasic decay curve. The shorter the turning point time is, the more photoholes the water oxidation employs effectively.

Supplementary Fig. 36 | Bias-dependent TAS for Ta:Fe₂O₃, Ta:Fe₂O₃@Fe₂O₃, and NiFe(OH)_x/Ta:Fe₂O₃@Fe₂O₃ photoanodes in 1M NaOH. Bias-dependent TAS results display that the long-lived holes appear at lower bias (0.7–0.8 V_{RHE}) in homojunction photoanode than that of the Ta:Fe₂O₃, in accordance with the negative shift of onset potential. However, no long-lived holes were observed at lower bias (0.7–0.8 V_{RHE}) after NiFe(OH)_x modification. This result may be related to two reasons. One is the hole transfer from homojunction to NiFe(OH)_x. The other is the hole absorption spectra in NiFeOOH overlapped with the photoholes in Fe₂O₃ (the reactive center in NiFeOOH electrocatalyst is Fe-site), which cannot be distinguished from the hole in hematite.¹³ Accordingly, at applied bias of 1.3 V_{RHE} (Fig. 6d), Ta:Fe₂O₃@Fe₂O₃/NiFe(OH)_x photoanode shows similar dynamics with Ta:Fe₂O₃@Fe₂O₃ at ms – s timescale probably due to the similar absorption coefficient of NiFe(OH)_x with Fe₂O₃.¹⁴

Comment 4): Overall, although the characterization of the structure and crystallinity of the material is very comprehensive, the fact that the performance is not a benchmark in the field requires bigger efforts on the PEC characterization to really differentiate this work and provide unique insights on what is the role of each treatment and how it improves the response. For all these reasons, given the high standards for publication in nature communication I am not able to accept for publication as it is, but dedicating some time to improve the PEC response could certainly improve the novelty of the paper.

Response 4): We agree with the reviewer that the PEC performance of our hematite photoanode is not the highest benchmark result, although it is still quite superior or comparable to most of the results reported so far. However, we believe it is more important for this research field to develop new concepts and mechanisms to improve the performance of hematite photoanodes, rather than numerical performance itself. Thus, the novelty of our work includes: **i) *in-situ* doping strategy with near “saturated” amount of Ta dopants; ii) the unique heat treatment of hybrid microwave annealing (HMA), iii) the partial and radial gradient nanorods with the gradient descent towards outside, iv) the pronounced effects on built-in electric field, and v) transient/time-resolved spectroscopic studies to reveal the kinetic mechanism.** In stark contrast to other reports of axial/radial gradient doping for the whole nanorods, our radial gradient doping region only exists from the core/shell interface to the shell part rather than the entire cross section of the nanorod. Consequently, the bulk of the central core part remains heavily Ta⁵⁺ doped, which provides high conductivity of homojunction nanorods. Moreover, the radial gradient that descends towards outside can induce less recombination near surface, improve spatial distribution of electric field, and enlarge the space charge region

According to the valuable comments of the reviewer, we have performed a substantial amount of additional experiments, which have significantly improved the quality of this manuscript. Now can emphasize more clearly the core essence of this work - the gradient Ta-doped homojunction nanorod results in high conductivity inside (heavy Ta⁵⁺ doping) while the surface states outside are passivated by the removal of surface defects caused by heavy Ta⁵⁺ doping. More crucially, this provides an additional electric field to

suppress charge recombination, leading to a significant enhancement of photocurrents as well as turn-on voltage simultaneously. Besides, homojunction enables NiFe(OH)_x cocatalyst to perform much better than on Ta:Fe₂O₃ photoanode, demonstrating the power of combining multiple strategies of high doping, homojunction and cocatalyst loading to improve the performance of a photoanode.

These results essentially provide new critical insights for designing efficient photoelectrodes for PEC water splitting with metal oxide semiconductors of poor optoelectronic properties by combining multiple strategies. We do think the work reported in this manuscript represents a significant progress (such as the novel nanostructure design, the synergetic strategy, and the distinctive PEC and spectroscopic responses) for the development of efficient photoanodes in the field of PEC water splitting, which are worth to be communicated in high impact journal of *Nature Communications*.

Reviewer #2

Comments to the Author: *The present manuscript reports on gradient tantalum doped hematite photoanodes modified with a NiFe(OH)_x co-catalyst (NiFe(OH)_x/Ta:Fe₂O₃@Fe₂O₃) showing optimized photocurrent density of 3.22 mA cm⁻² at 1.23 V RHE and onset potential of 0.55 V_{RHE} under simulated solar light illumination. The hematite nanorods were formed by hybrid microwave annealing (HMA), a method introduced by the authors in previous work. In my view, in spite of the significantly enhanced PEC performance of NiFe(OH)_x/Ta:Fe₂O₃@Fe₂O₃, the present manuscript requires revision, i.e. more detailed information, according to the points below for publication in Nature Communication:*

Comment 1): *There are various published papers about gradient doping (e.g. Chem. Sci., 2017, 8, 91-100; ChemSusChem, 2018, 11, 1873-1879; J. POWER SOURCES, 2020, 449, 227473; Phys. Chem. Chem. Phys., 2016, 18, 32735-32743). The effects of gradient doping described in the present manuscript are mainly an increased J_{ph} and a cathodic*

shift of V_{on} which are very similar to what already reported. However, the authors find for Ta:Fe₂O₃@Fe₂O₃ a much higher performance than previous studies. Hence, the reviewer wonders what makes the gradient Ta ion doping here presented unique compared with other doping strategies (Ti, Sn, Pt or P ion doping) previously reported. It is somehow hard to accept such a significantly enhanced PEC performance based on the present plain explanation.

Response 1): The most significant differences of our gradient doping strategy compared to those reported in literature by conventional thermal annealing (CTA) may lie in the following aspects: **i) the unique hybrid microwave annealing (HMA) treatment, ii) the *in-situ* doping strategy with near “saturated” amount of Ta dopants, iii) the partial and radial gradient nanorod with gradient descent towards outside, iv) generation of pronounced effects on built-in electric field, and v) the distinctive PEC and spectroscopic responses** All these merits led to a significant increase of J_{ph} and cathodic shift of V_{on} simultaneously.

As an alternative heating method to CTA, HMA has already been demonstrated to possess powerful potentials in fabrication of more efficient photoelectrodes owing to its unique characteristics - to achieve high temperature of 700~1000 °C in an extremely short time (2–3 min), to preserve the original nanostructure of the photoelectrode films, to generate high crystallinity and purity, and to realize negligible damage of FTO substrates. These distinctive features of HMA are highly desirable for the fabrication of high efficiency photoelectrodes. Particularly, the extreme condition of HMA could induce extraordinarily chemical reactions to form high temperature phase materials (e.g., FeNbO₄ and FeTaO₄) as reported in our previous papers (*ACS Catal.* **2019**, 9, 1289; *Adv. Funct. Mater.* **2019**, 29, 1805737). In comparison with CTA, HMA displays strong potential to boost much higher performance of metal oxide photoelectrodes (*Acc. Chem. Res.* **2019**, 52, 3132).

Ta dopants were *in-situ* doped into FeOOH lattices during the synthesis process. The dopant amount was carefully controlled to just “saturate” the well-grown FeOOH nanorods. Beyond this amount, the nanorod morphology would change into big

nanoparticles (Supplementary Fig. 2g, h). During conversion of FeOOH into Fe₂O₃ by HMA, the “saturated” Ta atoms would partially diffuse into the shell part to form gradient Ta doped core-shell homojunction. In stark contrast to other reports of axial/radial gradient doping for the whole nanorods, **our radial gradient doping region exists only from the core/shell interface to the shell part rather than the entire cross section of the nanorod. At the same time, the bulk of the centric core part remains heavily doped with Ta⁵⁺ ions, which provides the high conductivity of homojunction nanorods. Moreover, the radial gradient descends towards outside.** Actually, the large cation dopants prefer to occupy substitutional rather than interstitial sites (*Nano Lett.* **2011**, 11, 1775). In view of ionic radii (refer to Ionic radius - Wikipedia, Sn⁴⁺ = 83 pm, Fe³⁺ = 78.5 pm, Ta⁵⁺ = 78 pm, Pt⁴⁺ = 76.5 pm, Ti⁴⁺ = 74.5 pm), Ta⁵⁺ is quite close to Fe³⁺, which would not result in obvious lattice contraction (much smaller) or expansion (much bigger) once it is doped into hematite, which is beneficial for achieving heavy doping and high crystallinity simultaneously. **More importantly, high oxidation state Ta⁵⁺ ions gives an n-doping, which generates more excess electrons to enhance conductivity significantly compared to M⁴⁺ (M = Ti, Sn, Pt) dopants for the same doping amount.** Additionally, in view of crystal face stability (For bonding energies, please refer to *Lange's Handbook of Chemistry_15th Edition by John A. Dean*, $D^{\circ}_{298}/\text{kJ mol}^{-1} = \text{Ta-O} (805) > \text{Ti-O} (662) > \text{Sn-O} (548) > \text{Fe-O} (409) > \text{Pt-O} (347)$), the substitution of Ta for Fe in the hematite lattice might lower the surface energy of {001} facets owing to its high bonding energy (almost doubles that of Fe-O), making them more stable (*J. Phys. Chem. C* **2014**, 118, 16842). Consequently, Ta dopants would exert a more positive influence on the growth rate of {001} facets. The conductivity along the {001} facets is up to four orders of magnitude larger than that of the [110] direction (*ACS Nano* **2015**, 9, 7113; *J. Chem. Phys.* **2005**, 122, 144305). The nonmetallic phosphor doping into hematite is a completely different case, which tends to form FePO₄ locally (*Chem. Sci.* **2017**, 8, 91; *Energy Environ. Sci.* **2015**, 8, 1231). The structural stability and long-term durability might be its limitation with respect to these metal ion doping. However, it is difficult to make a conclusion which one (metallic or nonmetallic) would be better.

Another distinction of our gradient doping is that **the effect of built-in electric field is clearly displayed by the measurements of PEIS, charge separation/injection**

efficiency, and TAS, especially at lower potentials. These features were never experimentally proved in the literature for gradient doping photoelectrodes.

We have added the control experiments using CTA to make a clear comparison with HMA samples. We have also added bare “homojunction” ($\text{Fe}_2\text{O}_3@\text{Fe}_2\text{O}_3$) both by CTA and HMA and those with/without $\text{NiFe}(\text{OH})_x$ cocatalyst. Related revision has been made as highlighted.

(Line 13, Page 12)

More importantly, $\text{NiFe}(\text{OH})_x$ cocatalyst and homojunction formation have synergistic effects each other in improving PEC performance of Ta: Fe_2O_3 . It should be noted that all the photoanodes (with/without $\text{NiFe}(\text{OH})_x$ overlayer) prepared by conventional thermal annealing (CTA) show very inferior performance (Supplementary Fig. 16), which demonstrates that HMA is indispensable for high efficiency photoanodes. However, even CTA homojunction (Ta: $\text{Fe}_2\text{O}_3@\text{Fe}_2\text{O}_3$) does exhibit the beneficial effect of built-in electric field (modest V_{on} shift and J_{ph} increase, Supplementary Fig. 16b) relative to bare hematite homojunction ($\text{Fe}_2\text{O}_3@\text{Fe}_2\text{O}_3$) prepared by either HMA or CTA (only J_{ph} increases, Supplementary Fig. 17), indicating that the core part with a sufficient amount of Ta dopants is prerequisite to construct an effective homojunction so that the generated built-in electric field would promote charge separation efficiently. Interestingly, the superior quality of electrodes prepared by HMA is reflected not only in their own behavior but also in their performance after loading $\text{NiFe}(\text{OH})_x$ cocatalyst. This could be related to several issues of CTA, *i.e.* poor nanorod conductivity due to a limited amount of Sn dopant from FTO, evolved morphology, decreased surface area (Supplementary Fig. 18), and damage of FTO conductivity.

Supplementary Fig. 16 | *J-V* curves of the photoelectrodes prepared by CTA. From *J-V* curves (a), the optimization condition for Ta:Fe₂O₃ photoanode is 800°C for 20 min. Higher or lower temperature would obviously decrease the photocurrent density. CTA homojunction shows an enhanced J_{ph} and a cathodic shift of V_{on} relative to CTA Ta:Fe₂O₃ (b), which indicates the existence of built-in electric field in CTA homojunction as well (Further study is needed to confirm whether it is also a gradient homojunction). However, all the photoelectrodes by CTA with/without NiFe(OH)_x cocatalyst (1.26, 1.51, 1.64, and 1.91 mA cm⁻² for Ta:Fe₂O₃, Ta:Fe₂O₃@Fe₂O₃, NiFe(OH)_x/Ta:Fe₂O₃, NiFe(OH)_x/Ta:Fe₂O₃@Fe₂O₃, respectively) are inferior to the counterparts by HMA (1.93, 2.45, 2.48, and 3.22 mA cm⁻² for Ta:Fe₂O₃, Ta:Fe₂O₃@Fe₂O₃, NiFe(OH)_x/Ta:Fe₂O₃, NiFe(OH)_x/Ta:Fe₂O₃@Fe₂O₃, respectively). The distinctive features of HMA are highly desirable for fabrication of high efficiency photoanodes, which cannot be achieved by CTA.

Supplementary Fig. 17| Bare hematite with/without second growth prepared by CTA and HMA. For both CTA (a) and HMA (b) samples, Fe₂O₃@Fe₂O₃ shows a limited increase of J_{ph} and a negligible V_{on} shift relative to bare hematite (J_{ph} is slightly better for HMA), which demonstrates that the built-in electric field does not occur in this homojunction. Therefore, the sufficient amount of dopants (M^{n+} , $n>3$) in the core part is essential to construct an effective homojunction, in which built-in electric field would promote charge separation efficiently. Note that NiFe(OH)_x cocatalyst overlayer on all the HMA photoanodes promotes higher photocurrent density than on the CTA photoanodes, which demonstrates the high quality of electrodes prepared by HMA is reflected not only in their own behavior but also in the performance with cocatalyst loading. XRD (c) shows the similar pattern and intensity of peaks except the stronger intensity of (110) peak for homojunction, indicating the a preferential growth for bare hematite compared to Ta:Fe₂O₃. The LHE (d) of homojunction shows a slight increase relative to bare hematite, suggesting its negligible contribution to light absorption.

Supplementary Fig. 18 SEM images for the corresponding CTA homojunction ($\text{Ta:Fe}_2\text{O}_3@\text{Fe}_2\text{O}_3$) annealed at 800 °C for 20 min. Clearly, the diameter of nanorods becomes bigger and nanorod tips turn to be blunt, which would affect surface area, conductivity, and light harvesting ability of nanorods, resulting in the poor performance. Moreover, the undesirable factors accompanying CTA also restrain the effect of $\text{NiFe}(\text{OH})_x$ cocatalyst on the photoanodes, leading to a limited improvement of J_{ph} .

Comment 2): The authors should add the etching rate used for the TOF-SIMS measurements and for the depth profiling XPS analysis; this information is very important to verify the gradient doping (also sputter artifacts should be excluded).

Response 2): We thank the reviewer for this valuable comment. The depth etching rates (both XPS and TOF-SIMS) are highly dependent on the material properties including density, hardness, intension and conductivity. It is difficult to know the exact etching rate just from the estimation of etching time. However, the measured samples both for XPS and TOF-SIMS depth profiling were saved, which have been measured by the surface profiler (KLA Tencor P6 surface profilometer) to estimate the corresponding etched depth. The XPS sample shows the estimated depth of 200~250 nm (Supplementary Fig. 10a), which corresponds to the etching rate of 6~7.5 nm/min. The estimated thickness of the shell part is about 30~37.5 nm, which is also close to the observed SEM and TEM results. For TOF-SIMS sample, the etching rate is about 7.5~9 nm/min according to the estimated depth of 250~300 nm (Supplementary Fig. 10b). Accordingly, the thickness of the shell part is estimated to be 25~30 nm, which is consistent with the observed results of SEM and TEM. There should not exist sputter artifacts according to the experience of professional operators. The following statement has been added as highlighted.

(Second last line, Page 9)

Moreover, the calculated shell thicknesses are about 25~30 nm and 30~37.5 nm by the estimated etching rate of XPS (6~7.5 nm/min) and TOF-SIMS (7.5~9 nm/min), respectively (Supplementary Fig. 10). These results of XPS depth profile and TOF-SIMS are consistent with HRSTEM results, demonstrating the formation of gradient Ta-doped hematite homojunction.

(Line 2, Page 26)

Time-of-flight secondary ion mass spectrometry (TOF-SIMS) was carried out on a TOF-SIMS V instrument (ION-TOF GmbH, Germany) using a 10 keV Bi⁺ primary ion beam source. The etched depths were measured with a surface profiler (KLA Tencor P6 Profilometer).

Supplementary Fig. 10 Surface profiler profiles of XPS (a) and TOF-SIMS (b) samples. The etched depths of XPS and TOF-SIMS samples are estimated to be 200~250 nm and 250~300 nm, which corresponds to the etching rates of 6~7.5 nm/min and 7.5~9 nm/min, respectively.

Comment 3): The authors should add the results of electrochemical measurements of pristine hematite as a reference. Particularly important is the comparison of Mott-Schottky data of NiFe(OH)_x/Ta:Fe₂O₃@Fe₂O₃ vs. pristine hematite.

Response 3): We thank the reviewer for this comment. In the Response 1 to the Comment 1 of Reviewer #2, we compared bare photoanodes (including bare Fe₂O₃,

$\text{Fe}_2\text{O}_3@\text{Fe}_2\text{O}_3$, $\text{NiFe}(\text{OH})_x/\text{Fe}_2\text{O}_3$, and $\text{NiFe}(\text{OH})_x/\text{Fe}_2\text{O}_3@\text{Fe}_2\text{O}_3$ with no Ta doping) prepared by both CTA and HMA in Supplementary Fig. 17a,b. Besides, we also compared Ta-doped photoanodes (including $\text{Ta}:\text{Fe}_2\text{O}_3$, $\text{Ta}:\text{Fe}_2\text{O}_3@\text{Fe}_2\text{O}_3$, $\text{NiFe}(\text{OH})_x/\text{Ta}:\text{Fe}_2\text{O}_3$, and $\text{NiFe}(\text{OH})_x/\text{Ta}:\text{Fe}_2\text{O}_3@\text{Fe}_2\text{O}_3$) prepared by CTA in Supplementary Fig. 16. Hence, we think that the other extensive electrochemical measurements for bare hematite photoanode like what we did for $\text{Ta}:\text{Fe}_2\text{O}_3$ photoanodes are not necessary. The reasons include:

- i) The starting point of this work is $\text{Ta}:\text{Fe}_2\text{O}_3$ instead of bare hematite because of the “double-edged sword” effect by metal cation doping. This work attempts to eliminate bad and retain good effect of metal cation doping, which has been successfully done by the gradient Ta-doped homojunction. Based on $\text{Ta}:\text{Fe}_2\text{O}_3$ photoanode, this work devotes to further improving the already-achieved PEC performance of $\text{Ta}:\text{Fe}_2\text{O}_3$ and to deciphering the reasons for the improved performance by each applied strategy of homojunction formation and cocatalyst loading. The key point of this work is the formation of gradient Ta-doped homojunction and the base reference is $\text{Ta}:\text{Fe}_2\text{O}_3$ photoanode instead of bare hematite.
- ii) Extensive studies have already carried out on the effects of metal/nonmetal doping by theories and experimental measurements with bare hematite as a base reference. In the limited experiments we performed on bare and doped Fe_2O_3 , we could not find any significant deviation from those well-understood metal doping effects. Hence, this work focusses on the effect of the gradient Ta-doped homojunction with $\text{Ta}:\text{Fe}_2\text{O}_3$ photoanode as a base case.

Nevertheless, following the reviewer’s comment, we have added the Mott-Schottky plots of bare hematite and $\text{NiFe}(\text{OH})_x/\text{Ta}:\text{Fe}_2\text{O}_3@\text{Fe}_2\text{O}_3$, $\text{Ta}:\text{Fe}_2\text{O}_3$, and $\text{Ta}:\text{Fe}_2\text{O}_3@\text{Fe}_2\text{O}_3$ homojunction (Supplementary Fig. 19). We agree with the reviewer that this is the most important parameter to look at to see the effect of doping.

Supplementary Fig. 19 | Mott-Schottky plots of bare Fe_2O_3 , $\text{Ta}:\text{Fe}_2\text{O}_3$, $\text{Ta}:\text{Fe}_2\text{O}_3@\text{Fe}_2\text{O}_3$, and $\text{NiFe}(\text{OH})_x/\text{Ta}:\text{Fe}_2\text{O}_3@\text{Fe}_2\text{O}_3$ photoanodes. Mott-Schottky plot can provide the flat band potential (E_{FB} , the intercept value on the X-axis), and the donor density (N_{D} , inversely proportional to the slope). Clearly, $\text{Ta}:\text{Fe}_2\text{O}_3$ shows a remarkably decreased slope compared to that of bare Fe_2O_3 , indicating a significant increase of donor density to improve the poor conductivity of Fe_2O_3 . On the other hand, E_{FB} of $\text{Ta}:\text{Fe}_2\text{O}_3$ shows a positive shift of ~ 50 mV, leading to unwanted increased overpotential (a side effect of doping). In stark contrast, homojunction formation creates a smaller slope and a lower E_{FB} relative to those of $\text{Ta}:\text{Fe}_2\text{O}_3$, which addresses both poor conductivity and high overpotential simultaneously, consistent with its higher photocurrent and lower V_{on} . Generally, the addition of an extremely thin cocatalyst layer should not affect the doping level or carrier density within hematite electrode. The presented N_{D} and E_{FB} of $\text{NiFe}(\text{OH})_x/\text{Ta}:\text{Fe}_2\text{O}_3@\text{Fe}_2\text{O}_3$ are quite similar to those of homojunction, which are also in agreement with the previous reports of cocatalyst layer modification.^{5, 6} It should be noted that Grätzel and coworkers have recently demonstrated through approximation of a flat surface and careful calculation of active surface area that an extremely thin cocatalyst layer (CoFeO_x) on hematite shifts only the photocurrent while leaving E_{FB} unchanged.^{7, 8}

Refs:

5. Li M, *et al.* Zipping up NiFe(OH)_x-encapsulated hematite to achieve an ultralow turn-on potential for water oxidation. *ACS Energy Lett* **4**, 1983-1990 (2019).
6. Malara F, *et al.* α -Fe₂O₃/NiOOH: An Effective Heterostructure for Photoelectrochemical Water Oxidation. *ACS Catal* **5**, 5292-5300 (2015).
7. Cesar I, Sivula K, Kay A, Zboril R, Grätzel M. Influence of Feature Size, Film Thickness, and Silicon Doping on the Performance of Nanostructured Hematite Photoanodes for Solar Water Splitting. *J Phys Chem C* **113**, 772-782 (2009).
8. Liardet L, Katz JE, Luo J, Grätzel M, Hu X. An ultrathin cobalt–iron oxide catalyst for water oxidation on nanostructured hematite photoanodes. *J Mater Chem A* **7**, 6012-6020 (2019).

Comment 4): We strongly suggest the authors to apply a mask in front of the photoanodes for the PEC measurements, this to define the activate area in hematite and to prevent any influences from light scattering which may enhance the PEC performance. Furthermore, we would kindly ask the authors to provide a picture of their PEC measurement system. Overall, I recommend this manuscript to be strongly revised before considering publication in Nature Communications.

Response 4): Thanks for the reviewer's suggestion and we are sorry for this misunderstanding. The reviewer probably noticed the synthesized photoelectrodes without a mask in our previous report (*Adv. Funct. Mater.* **2019**, 29, 1805737). In fact, we examined the synthesized photoelectrodes with and without mask because of the same concern from a reviewer of our previous publication. Indeed, the electrode without mask showed an enhanced PEC performance, but to a very limited extent, which did not affect the validity of our results at all. Nevertheless, we have been using the masked photoelectrodes since then including this work. Actually in Methods section of this manuscript, you can find the statement that the **well-cut samples (12.5×12.5 mm²)** were put on the slightly compacted graphite powder (60 mL) as a susceptor... and **note that the exposed surface area of all the prepared photoanodes was geometric area (1 cm × 1 cm)** and the electrolyte was slowly stirred during the measurements....., which also can be found in our latest papers (*ACS Catal.* **2019**, 9, 1289; *Chem. Sci.* **2019**, 10, 10436).

To make the situation much clearer, we have revised this description and also provided a picture of PEC measurement system (Supplementary Fig. 12) in the revised manuscript.

(Line 5, Page 25)

Note that the exposed surface area of all the prepared photoanodes was geometric area ($1\text{ cm} \times 1\text{ cm}$, other parts including the edges were covered with epoxy resin) and the electrolyte was slowly stirred during the measurements...

Supplementary Fig. 12| A typical PEC system used in this study.

Reviewer #3

Comments to the Author: This work reported a core-shell formation of gradient tantalum-doped hematite homojunction nanorods that improves photocurrents and turn-on voltage for photoelectrochemical water splitting. The authors claimed that homojunction formation could passivate the surface states and provide an electric field to suppress charge recombination, leading to the enhanced photocurrents and turn-on voltage. Besides, their homojunction presented a synergy with $\text{NiFe}(\text{OH})_x$ cocatalysts with additional improvements of photocurrents and turn-on voltage. This work is novel and provides some insights for designing efficient semiconductor photoelectrodes for photoelectrochemical water splitting, which could be recommended for a publication in *Nature Communications* after addressing the following comments.

Comment 1): What is the difference between tantalum-doped hematite vs. bare hematite? Provide photoelectrochemical performance and other data of Fe_2O_3 and $\text{Fe}_2\text{O}_3@\text{Fe}_2\text{O}_3$.

Response 1): The bare hematite homojunction ($\text{Fe}_2\text{O}_3@\text{Fe}_2\text{O}_3$) prepared by both HMA and CTA does not show the effect of built-in electric field compared with that of Ta-doped $\text{Ta}:\text{Fe}_2\text{O}_3@\text{Fe}_2\text{O}_3$ homojunction and gives inferior PEC performance. We have added the reference photoanodes of bare hematite with and without $\text{NiFe}(\text{OH})_x$ cocatalyst layer, including the samples by both annealing methods of CTA and HMA (Supplementary Fig. 17a,b). The corresponding XRD and LHE data of Fe_2O_3 and $\text{Fe}_2\text{O}_3@\text{Fe}_2\text{O}_3$ have been added in Supplementary Fig. 17c, d in the revised manuscript.

Supplementary Fig. 17| Bare hematite with/without second growth prepared by CTA and HMA. For both CTA (a) and HMA (b) samples, $\text{Fe}_2\text{O}_3@\text{Fe}_2\text{O}_3$ shows a limited increase of J_{ph} and a negligible V_{on} shift relative to bare hematite (J_{ph} is slightly better for HMA), which demonstrates that the built-in electric field does not occur in this homojunction. Therefore, the sufficient amount of dopants (M^{n+} , $n>3$) in the core part is essential to construct an effective homojunction, in which built-in electric field would

promote charge separation efficiently. Note that $\text{NiFe}(\text{OH})_x$ cocatalyst overlayer on all the HMA photoanodes promotes higher photocurrent density than on the CTA photoanodes, which demonstrates the high quality of electrodes prepared by HMA is reflected not only in their own behavior but also in the performance with cocatalyst loading. XRD (c) shows the similar pattern and intensity of peaks except the stronger intensity of (110) peak for homojunction, indicating the a preferential growth for bare hematite compared to $\text{Ta}:\text{Fe}_2\text{O}_3$. The LHE (d) of homojunction shows a slight increase relative to bare hematite, suggesting its negligible contribution to light absorption.

Comment 2): What are the states of Ta atoms in the $\text{Ta}:\text{Fe}_2\text{O}_3$ and $\text{Ta}:\text{Fe}_2\text{O}_3@\text{Fe}_2\text{O}_3$, are they isolated atoms or nanoclusters? Only HRSTEM data is not sufficient.

Response 2): We thank the reviewer for this question. We have added synchrotron PXRD results in the revised manuscript. Powder samples of $\text{Ta}:\text{Fe}_2\text{O}_3$ and $\text{Ta}:\text{Fe}_2\text{O}_3@\text{Fe}_2\text{O}_3$ (scratched from the corresponding film ones) were measured by PXRD. The tunable, monochromatic and brilliance of the high energy beam source of synchrotron radiation made it possible to collect XRD features of structural changes even the small concentration of element doping.

(Line 4, Page 10)

To further clarify the state of Ta in the bulk nanorods (isolated atoms or nanoclusters), $\text{Ta}:\text{Fe}_2\text{O}_3$ and $\text{Ta}:\text{Fe}_2\text{O}_3@\text{Fe}_2\text{O}_3$ were analyzed by synchrotron powder XRD (PXRD) using powder samples scratched from the corresponding photoanode films. Both samples show exactly the same PXRD patterns (Supplementary Fig. 11a), only involving two phases of Fe_2O_3 and SnO_2 (scratched off from FTO substrate). The Rietveld refinement profile of $\text{Ta}:\text{Fe}_2\text{O}_3@\text{Fe}_2\text{O}_3$ was obtained by Full Prof Suite software (Supplementary Fig. 11b, c). All the peaks were completely assignable to Fe_2O_3 (R -3c) and SnO_2 (P 42/m n m). No relevant phases of Ta metal or oxides (such as TaO, Ta_2O_3 , TaO_2 and Ta_2O_5) were found. Note that the extreme conditions provided by HMA are more favorable to form high-crystallinity materials relative to CTA. Hence, the nanoclusters (if existed) in the synthesized samples should be crystalline as well, which would have been detected by synchrotron PXRD. Combining synchrotron PXRD with HRSTEM results, we can

conclude that the state of Ta in both samples should be isolated and randomly dispersed atoms.

Supplementary Fig. 11| Synchrotron PXRD ($\lambda=0.65303 \text{ \AA}$). Synchrotron PXRD patterns of Ta: Fe_2O_3 and Ta: Fe_2O_3 @ Fe_2O_3 (a). Rietveld refinement profile of

Ta:Fe₂O₃@Fe₂O₃ by the Full Prof Suite software (b). Parameter table of the Rietveld refinement (c). Here, Yobs, Ycalc, Yobs-Ycalc, Bragg_position, R_p, R_{wp}, R_{exp}, S represent the experimental data, the calculated data, the difference of experimental and calculated data, Bragg's position, the profile factor, the weighted profile R factor, the expected R factor, goodness of fit, respectively. Note that lattice parameters, isotropic strain broadening, asymmetry, isotropic size broadening, and GauSiz are considered for the reliable profile matching.

Comment 3): There is only one STEM image of Ta:Fe₂O₃ and Ta:Fe₂O₃@Fe₂O₃ in the paper, provide more STEM data on other Ta:Fe₂O₃ and Ta:Fe₂O₃@Fe₂O₃ to see the reproducibility.

Response 3): We have added more STEM data of Ta:Fe₂O₃ and Ta:Fe₂O₃@Fe₂O₃ in the revised manuscript (Supplementary Fig. 5). Revised text as highlighted:

(Line 14, Page 6)

The structure of Ta:Fe₂O₃@Fe₂O₃ homojunction nanorods and Ta:Fe₂O₃ were further examined by scanning transmission electron microscopy (STEM). The high-angle annular dark-field STEM (HAADF-STEM) image of Ta:Fe₂O₃ in Fig. 2a (more STEM images in Supplementary Fig. 5) clearly shows Ta atoms in the α-Fe₂O₃ lattices (circled), demonstrating that *in-situ* doping of Ta into FeOOH was successful.

Supplementary Fig. 5 | STEM images for Ta:Fe₂O₃ and Ta:Fe₂O₃@Fe₂O₃. (a) HRSTEM image of Ta:Fe₂O₃ nanorods (circles denote Ta atoms). (b, c) STEM images of Ta:Fe₂O₃@Fe₂O₃ homojunction nanorods. (d) HRSTEM of the homojunction nanorod. The scale bar is 5 nm in (a, d), 50 nm in (b) and 30 nm in (c).

Comment 4): Page 10, Figure 3a, why are there no Ta peaks in the XRD patterns?

Response 4): Appearance of the related Ta peaks in XRD pattern indicates two possibilities. First, some Ta atoms did not go into hematite lattices but form separate metallic Ta/oxide particles on the surface of nanorods. The other possibility is that some Ta nanoclusters are formed inside the nanorods. In our case, the controlled amount of Ta atoms was incorporated essentially completely into the hematite lattices to occupy substitutional sites (Fig. 2g) without changing the structure of hematite (Fig. 2b). In addition, no attached nanoparticles were observed on the surface of nanorods (Fig. 2a, e and Supplementary Fig. 5a-c). Moreover, synchrotron PXRD (Supplementary Fig. 11) confirms the absence of Ta related phases. Actually, no Ta related peaks in XRD pattern in turn demonstrate that all the involved Ta atoms have been successfully doped into hematite lattices and did not aggregate into nanoclusters.

Comment 5): *There are no physical characterizations of NiFe(OH)_x/Ta:Fe₂O₃ and NiFe(OH)_x/Ta:Fe₂O₃@Fe₂O₃, perform these measurements.*

Response 5): In fact, we had some physical characterizations of NiFe(OH)_x cocatalyst including SEM and UV-vis spectrometer in Supplementary Fig. 15d, e. In the revised manuscript, we have added XPS O 1s spectra (Supplementary Fig. 15f) and SEM-EDS (Supplementary Fig. 15g, h) to further characterize the cocatalyst. The our previous results demonstrate that we can apply this strategy to successfully deposit FeOOH cocatalyst on hematite and BiVO₄ photoanodes (*Adv. Funct. Mater.* **2019**, 29, 1805737; *Angew. Chem. Int. Ed.* **2018**, 57, 2248). Based on the results of XPS O 1s spectra and SEM-EDS, it is a reasonable to conclude that the cocatalyst is NiFe(OH)_x. The cocatalyst composition deposited on homojunction should be the same as that on Ta:Fe₂O₃

Supplementary Fig. 15| Optimization and characterization of NiFe(OH)_x cocatalyst. SEM images (a, 20 min. b, 60 min. c, 100 min.) and corresponding *J*–*V* curves (d) with different durations at 30 °C. Light absorption before and after optimal (60 min duration) cocatalyst modification (e). XPS O 1s spectra (f). SEM-EDS images (g, h). The nickel and iron chloride solution can be gradually hydrolyzed to form NiFe(OH)_x in a dilute concentration. From SEM images (barely taken without any sputtering of noble metal), 20 min duration (a) gives an invisible deposition, but shows an improved *J*_{ph} (2.84 mA cm⁻²). With 60 min duration, however, a thin and uniform layer of NiFe(OH)_x was deposited (b) because some small nanoparticles can be seen. Besides, the poor contrast of image relative to the previous one also indicates the successful deposition of NiFe(OH)_x, providing a maximum of *J*_{ph} (3.22 mA cm⁻²). On the other hand, this thin and uniform NiFe(OH)_x layer does not change any light absorption (e). When duration reaches 100 min, a great number of bigger nanoparticles can be observed clearly(c), which actually decreases the *J*_{ph} (2.63 mA cm⁻²) due to too much an amount. XPS O 1s spectra (f) shows

that the OH peak intensity of homojunction modified with cocatalyst becomes clearly higher than that of bare one. Besides, SEM-EDS (g, h) detects the signal of Ni element (~0.41wt%). Our previous results demonstrated that this strategy deposited FeOOH cocatalyst on hematite successfully. Based on these results, it is reasonable that the cocatalyst is NiFe(OH)_x.

Comment 6): Provide more details about the information of the photoelectrochemical cell.

Response 6): The detail information of PEC cell (Supplementary Fig. 12) has been added in the revised manuscript.

Supplementary Fig. 12| A typical PEC system used in this study.

Comment 7): Page 14, provide more details about how the H₂ and O₂ were measured. Run multiple sets of gas analysis experiments to get statistics.

Response 7): We have added the detail PEC cell about how H₂ and O₂ were measured (Supplementary Fig. 26) and also added the other two sets of gas analysis experiments (Fig. R2). The average values of the three data set were used to present the gas evolution curves in Supplementary Fig. 27. Related text has been revised as highlighted.

(Line 16, Page 28)

The amounts of H₂ and O₂ gases evolved from a closed circulation PEC system were analyzed by a gas chromatograph (GC, HP 7890) equipped with a thermal conductivity detector (TCD) and a packed column (Supelco, Carboxen 1000) using high purity Ar as a carrier gas (at a constant rate of 10 mL min⁻¹).

Supplementary Fig. 26| A typical PEC system used in this study with a closed circulation for the measurement of H₂ and O₂ evolution.

Fig. R2 Three sets of gas evolution measurements. First (a), second (b) and third (c) sets. The F.E.'s of O_2 evolution reaction with first, second, and third sets are 93 %, 94%, and 90 %, respectively. The corresponding F.E.'s for H_2 are 96%, 98%, and 93%, respectively.

Supplementary Fig. 27| Evolution of O_2 and H_2 over $\text{NiFe(OH)}_x/\text{Ta:Fe}_2\text{O}_3@/\text{Fe}_2\text{O}_3$ photoanode at a constant potential of $1.23 V_{\text{RHE}}$. Each point represents an average value from three sets of measurements. The ratio of evolved O_2 and H_2 is close to

stoichiometry, and F.E.'s of O₂ and H₂ evolution reaction is 92.3 % and 95.8 %, respectively.

Editorial Comments

In an effort to ensure reproducibility of research data, we now also require that you provide a separate source data file. The source data file should, as a minimum, contain the raw data underlying all reported averages in graphs and charts, and uncropped versions of any gels or blots presented in the figures. To learn more about our motivation behind this policy, please see <https://www.nature.com/articles/s41467-018-06012-8>.

Within the source data file, each figure or table (in the main manuscript and in the Supplementary Information) containing relevant data should be represented by a single sheet in an Excel document, or a single .txt file or other file type in a zipped folder. Blot and gel images should be pasted in and labelled with the relevant panel and identifying information such as the antibody used. We also encourage you to include any other types of raw data that may be appropriate. An example source data file is available demonstrating the correct format: <https://www.nature.com/documents/ncomms-example-source-data.xlsx>

The file should be labelled 'Source Data', with the title and a brief description included in your cover letter, and should be mentioned in all relevant figure legends using the template text below:

“Source data are provided as a Source Data file.”

Response: We have provided the source data file in excel and mentioned in all figure legends. Revision in highlight.

Data availability

The data that support the plots within this paper and other findings of this study are available from the corresponding author upon reasonable request. The source data

underlying Figs. 2g, 3, 4, 5 and 6 and Supplementary Figs. 2, 3a, 6, 7, 8, 10, 11, 13, 14, 15d-f, 16, 17, 19, 20, 22, 23, 24, 25, 27, 29, 30, 31, 32, 33, 34, 35 and 36 are provided as

a	Source	Data	file
---	--------	------	------

Reviewer #1 (Remarks to the Author):

The authors have addressed successfully the comments I indicated, and I consider it is now acceptable for publication.

Reviewer #2 (Remarks to the Author):

While the authors have revised their manuscript and provided additional data, I still think that the PEC setup used (as shown in fig. 12) remains a weak point.

The authors mention "...we have been using the masked photoelectrodes since then including this work". How exactly were the photo-electrodes masked? If the authors refer to the frame of the epoxy resin, probably this is not enough. A mask (other than the insulating resin) should be applied to to define the activate area in hematite and to prevent any influences from light scattering.

The picture in fig. 12 shows in fact that there is quite a lot of diffuse light in the setup, likely originating from the white background underneath the cell and from the cell glass walls.

A black background can be used (underneath and around the cell), to avoid diffuse light to bounce back to the photo-electrode.

In general, one wonders if this setup is designed well enough to avoid an overestimation of the PEC performance results.

I still think this point needs to be properly addressed before consider this manuscript for publication.

Reviewer #3 (Remarks to the Author):

The authors have satisfactorily addressed all my concerns. Their response to other reviewers' concerns/comments appear adequate to me, too. I, therefore, support accepting it for Nat. Common.

Response to Editor and Reviewers' Comments

We thank the editor and reviewers for their careful evaluation, comments and suggestions to greatly improve the quality of this work. We are pleased to see that Reviewer #1 and Reviewer #3 are satisfied with our responses for the first revision. We have made a point-by-point response to each additional comment raised by Reviewer #2 in the following.

Reviewer #2

Comments to the Author: While the authors have revised their manuscript and provided additional data, I still think that the PEC setup used (as shown in fig. 12) remains a weak point.

Comment 1): The authors mention "...we have been using the masked photoelectrodes since then including this work". How exactly were the photo-electrodes masked? If the authors refer to the frame of the epoxy resin, probably this is not enough. A mask (other than the insulating resin) should be applied to define the activate area in hematite and to prevent any influences from light scattering.

Response 1): We tried to make the active area of each photoanode be 1 cm × 1 cm. The exact active area was recalculated by the area pixels of the taken picture due to the slight fluctuation of the marked area. Although there is no actual mask (no strict sense of "active area"), the reproducibility and consistency were able to be kept well (the insulating and black resin actually has some mask function). Moreover, our PEC system is set up in a black box, preventing any enviromental influence (Figure R1). Compared with some other groups' set up in the air (no protection), ours should be much better.

Light scattering might exist in the glass cell, but our PEC measurements were done in a completely closed system. All the measured photoelectrodes would experience exactly the same conditions. The control experiments show a very limited effect by light scattering (as mentioned below). The main thrust of this work includes the novel gradient homojunction nanostructure, the synergetic strategy, and the distinctive PEC and spectroscopic response. This work essentially provides new critical insights for designing

efficient photoelectrodes in the field of PEC water splitting although the measured performance is not a benchmark one. It should be emphasized that the exact value (a little lower or higher one) varied by different PEC systems (e.g., with/without a strict sense of mask, in the air/in the black box) would not affect the novelty and originality of this work.

Figure R1. The picture of PEC system in a black box. Note that the door of box would be completely closed during actual experiments.

Comment 2): The picture in fig. 12 shows in fact that there is quite a lot of diffuse light in the setup, likely originating from the white background underneath the cell and from the cell glass walls.

Response 2): We are sorry for the misunderstanding of this picture (Supplementary Fig. 12). In order to get a clear and bright picture of our PEC system, an underneath white paper and the flash mode of the camera was intentionally used when the picture was taken. Moreover, the contrast and brightness of the taken picture was also adjusted to display it very well. That is why the picture shows an obvious effect of strong scattering light (see Figure R2 in a normal mode). It should be noted that there is no underneath white paper during the actual experiments. Moreover, the bottom of photoelectrodes were

also covered by the black resin, which enables that the PEC performance would not be influenced by the back bounce of the diffused light (Figure R3).

The diffuse light from glass cell walls should be very limited because of its high transparency (Figure R4). Generally, a glass cell is worldwide used, especially for PEC measurements (*Nat. Commun.*, 2019, **10**, 5282; *Nat. Commun.*, 2016, **7**, 11943; *Sci. Adv.*, 2016, **2**, e1501764; *Nat. Commun.*, 2019, **10**, 1779; *Nat. Commun.*, 2019, **10**, 3388; *Nat. Commun.*, 2019, **10**, 3687).

Figure R2. The picture of PEC system was taken in a normal way (without flash mode and no adjustment of brightness and contrast).

Figure R3. The picture of photoanode was taken in a normal way (without flash mode and no adjustment of brightness and contrast). **a)**, Top side. **b)** Back side.

Figure R4. The picture of glass cell was taken in a normal way (without flash mode and no adjustment of brightness and contrast).

***Comment 3):** A black background can be used (underneath and around the cell), to avoid diffuse light to bounce back to the photo-electrode. In general, one wonders if this setup is designed well enough to avoid an overestimation of the PEC performance results. I still think this point needs to be properly addressed before consider this manuscript for publication.*

Response 3): We compared the measurements with and without black background (underneath and around the cell, Figure R5) according to reviewer's suggestion. The result shows that the diffuse light has a very limited influence (less than 5%, Figure R6). In conclusion, ours set up might not be a perfect one, but it produces consistent data sets among different samples.

Figure R5. The glass cell was fully encapsulated by the black plastic film.

Figure R6. Comparison of J - V curves with and without black ground of glass cell.

REVIEWERS' COMMENTS:

Reviewer #2 (Remarks to the Author):

The authors have revised their manuscript according to the reviewer's requests.